# Minimax Forward and Backward Learning of Evolving Tasks with Performance Guarantees

Verónica Álvarez[1]    Santiago Mazuelas[1,2]    Jose A. Lozano[1,3]

[1]Basque Center of Applied Mathematics (BCAM)

[2]IKERBASQUE-Basque Foundation for Science    [3]University of the Basque Country UPV/EHU

{valvarez, smazuelas, jlozano}@bcamath.org

## Abstract

For a sequence of classification tasks that arrive over time, it is common that tasks are evolving in the sense that consecutive tasks often have a higher similarity. The incremental learning of a growing sequence of tasks holds promise to enable accurate classification even with few samples per task by leveraging information from all the tasks in the sequence (forward and backward learning). However, existing techniques developed for continual learning and concept drift adaptation are either designed for tasks with time-independent similarities or only aim to learn the last task in the sequence. This paper presents incremental minimax risk classifiers (IMRCs) that effectively exploit forward and backward learning and account for evolving tasks. In addition, we analytically characterize the performance improvement provided by forward and backward learning in terms of the tasks' expected quadratic change and the number of tasks. The experimental evaluation shows that IMRCs can result in a significant performance improvement, especially for reduced sample sizes.

## 1   Introduction

In practical scenarios, it is often of interest to incrementally learn a growing sequence of classification problems (tasks) that arrive over time. In such a sequence, it is common that tasks are evolving in the sense that consecutive tasks often have a higher similarity. Examples of evolving tasks are the classification of portraits from different time periods [1] and the classification of spam emails over time [2]; in these problems, the similarity between consecutive tasks (portraits of consecutive time periods and emails from consecutive years) is significantly higher (see Figure 1). The incremental learning of a growing sequence of tasks holds promise to significantly improve performance by leveraging information from different tasks. Specifically, at each time step, information from preceding tasks can be used to improve the performance of the last task (forward learning) and, reciprocally, the information from the last task can be used to improve the performance of the preceding tasks (backward learning) [3–5]. Such transfer of information can enable accurate classification even in cases with reduced sample sizes, thus significantly increasing the effective sample size (ESS) of each task. However, exploiting the benefits of forward and backward learning is challenging due to the continuous arrival of samples from tasks characterized by different underlying distributions [6–8].

Techniques developed for concept drift adaptation (aka learning in a dynamic scenario) [9–16] are designed for evolving tasks but only aim to learn the last task in the sequence. In particular, methods based on learning rates learn the last task by slightly updating the classification rule for the preceding task [12, 13]; and methods based on sliding windows learn the last task by using a set of stored samples from the most recent preceding tasks [14, 15]. In concept drift adaptation, at each time

37th Conference on Neural Information Processing Systems (NeurIPS 2023).

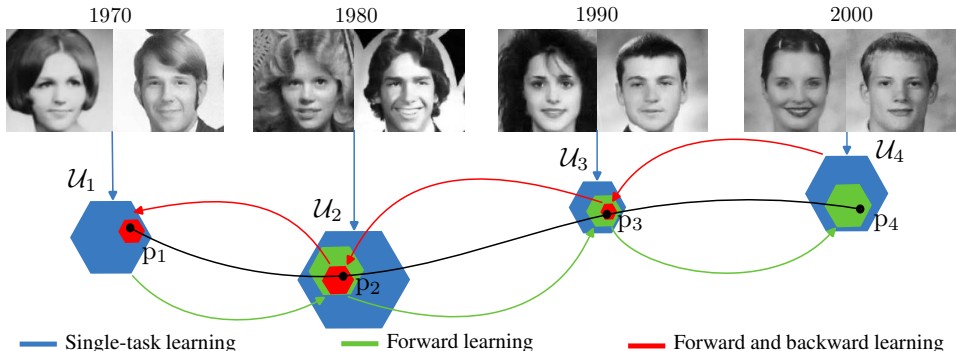

Figure 1: Tasks that arrive over time are characterized by different underlying distributions and consecutive tasks often have a higher similarity. The black line represents the evolution of the underlying distributions that characterize different tasks. IMRCs minimize the worst-case error probability over uncertainty sets $\mathcal{U}_i$ that can include the underlying distribution $p_i$. A single task uncertainty set (blue hexagons) can be obtained leveraging information only from the corresponding task, while a forward uncertainty set (green hexagons) can be obtained leveraging information from preceding tasks. Then, the proposed methodology obtains forward and backward uncertainty sets (red hexagons) leveraging information from all the tasks in the sequence.

step only the last task is considered of interest, and is learned leveraging information from the most recent preceding tasks since they are the most similar to the last task.

Techniques developed for continual learning (aka lifelong learning) [4–7, 17–20] aim to learn the whole sequence of tasks but existing methods are designed for situations where tasks' similarities do not depend on the time steps when tasks are observed. In particular, methods based on dynamic architectures learn shared parameters using samples from all the tasks together with task-specific parameters using samples from the corresponding task [6, 20]; and methods based on replay learn parameters using a pool of stored samples from all the preceding tasks together with the samples from the last task [5, 7]. Existing methods for continual learning are not designed for evolving tasks and consider scenarios in which the order of the tasks in the sequence is not relevant. In the current literature of continual learning, only [21] considers scenarios with evolving tasks but focus on the theoretical analysis of transferring information from the preceding tasks.

This paper presents incremental minimax risk classifiers (IMRCs) that determine classification rules minimizing worst-case error probabilities over uncertainty sets that can contain the sequence of evolving underlying distributions. The proposed techniques can effectively exploit forward and backward learning by obtaining uncertainty sets that get smaller using information from all the tasks (see Figure 1). Specifically, the main contributions presented in the paper are as follows.

- We propose forward learning techniques that recursively use the information from preceding tasks to reduce the uncertainty set of the last task.

- We propose forward and backward learning techniques that use the information from the last task to further reduce the sequence of uncertainty sets obtained with forward learning.

- We analytically characterize the increase in ESS provided by the presented methods in terms of the expected quadratic change between consecutive tasks and the number of tasks.

- We numerically quantify the performance improvement provided by IMRCs in comparison with existing techniques using multiple datasets, sample sizes, and number of tasks.

**Notations**  Calligraphic letters represent sets; $\|\cdot\|_1$ and $\|\cdot\|_\infty$ denote the 1-norm and the infinity norm of its argument, respectively; $\preceq$ and $\succeq$ denote vector inequalities; $\mathbb{I}\{\cdot\}$ denotes the indicator function; and $\mathbb{E}_p\{\cdot\}$ and $\mathbb{V}\mathrm{ar}_p\{\cdot\}$ denote the expectation and the variance of its argument with respect to distribution $p$. For a vector $\mathbf{v}$, $v^{(i)}$ and $\mathbf{v}^{\mathrm{T}}$ denote its i-th component and transpose. Non-linear operators acting on vectors denote component-wise operations. For instance, $|\mathbf{v}|$ and $\mathbf{v}^2$ denote the vector formed by the absolute value and the square of each component, respectively. For the reader's convenience, we also provide in Table 2 in Appendix A a list with the main notions used in the paper and their corresponding notations.

## 2 Preliminaries

This section first formulates the problem of incrementally learning a growing sequence of tasks, and describes evolving tasks in comparison with the time-independent assumption common in continual learning. Then, we briefly describe MRCs that determine classification rules by minimizing the worst-case error probability over an uncertainty set.

### 2.1 Problem formulation

In the following, we denote by $\mathcal{X}$ the set of instances or attributes, $\mathcal{Y}$ the set of labels or classes, $\Delta(\mathcal{X} \times \mathcal{Y})$ the set of probability distributions over $\mathcal{X} \times \mathcal{Y}$, and $\mathrm{T}(\mathcal{X}, \mathcal{Y})$ the set of classification rules. A classification task is characterized by an underlying distribution $\mathrm{p}^* \in \Delta(\mathcal{X} \times \mathcal{Y})$, and standard supervised classification methods use a sample set $D = \{(x_i, y_i)\}_{i=1}^n$ formed by $n$ i.i.d samples from distribution $\mathrm{p}^*$ to find a classification rule $\mathrm{h} \in \mathrm{T}(\mathcal{X}, \mathcal{Y})$ with small expected loss $\ell(\mathrm{h}, \mathrm{p}^*)$.

In the addressed settings, sample sets $D_1, D_2, \ldots$ arrive over time steps $1, 2, \ldots$ corresponding with different classification tasks characterized by underlying distributions $\mathrm{p}_1, \mathrm{p}_2, \ldots$. Incremental learning aims to continually learn over time the growing sequence of tasks exploiting information acquired from all the tasks. At each time step $k$, learning methods obtain classification rules $\mathrm{h}_1, \mathrm{h}_2, \ldots, \mathrm{h}_k$ for the current sequence of $k$ tasks using the new sample set $D_k$ and certain information retained from step $k-1$. The performance of learning methods is assessed at each time step $k$ by quantifying the performance in the current $k$ tasks. For instance, overall performance can be assessed by the averaged error $\frac{1}{k} \sum_{j=1}^k \ell(\mathrm{h}_j, \mathrm{p}_j)$ with $\ell(\mathrm{h}_j, \mathrm{p}_j)$ the expected loss of the classification rule $\mathrm{h}_j$ for distribution $\mathrm{p}_j$.

Evolving tasks are considered by methods for concept drift adaptation, but existing continual learning methods are designed for scenarios where tasks' similarities do not depend on the time steps when tasks are observed. These scenarios are usually mathematically modeled assuming that the tasks' distributions $\mathrm{p}_1, \mathrm{p}_2, \ldots$ are independent and identically distributed (i.i.d.) [18, 19]. In this paper, we develop techniques designed for evolving tasks that can be mathematically modeled assuming that the changes between consecutive distributions $\mathrm{p}_2 - \mathrm{p}_1, \mathrm{p}_3 - \mathrm{p}_2, \ldots$ are independent and zero-mean (evolving task assumption). The assumption in this paper is not stronger than the usual i.i.d. assumption and can better describe evolving tasks; their main difference is considering independent changes between consecutive distributions instead of independent distributions. Note that with independent distributions the difference between the distributions of the $i$-th and the $(i+t)$-th tasks has zero mean and variance $\mathbb{V}\mathrm{ar}\{\mathrm{p}_{i+t} - \mathrm{p}_i\} = \mathbb{V}\mathrm{ar}\{\mathrm{p}_{i+1} - \mathrm{p}_i\} = 2\mathbb{V}\mathrm{ar}\{\mathrm{p}_1\}$ that does not depend on $t$. On the other hand, with independent changes between consecutive distributions, the difference between the distributions of the $i$-th and the $(i+t)$-th tasks has zero mean and variance $\mathbb{V}\mathrm{ar}\{\mathrm{p}_{i+t} - \mathrm{p}_i\} = \sum_{j=1}^t \mathbb{V}\mathrm{ar}\{\mathrm{p}_{i+j} - \mathrm{p}_{i+j-1}\}$ that increases with $t$. Appendix H further describes how the assumption used in the paper is more appropriate for evolving tasks than the conventional i.i.d. assumption using real datasets.

### 2.2 Minimax risk classifiers

The methods presented in the paper are based on robust risk minimization [22, 23] instead of empirical risk minimization since training samples corresponding to different tasks follow different distributions. In particular, we utilize MRCs [24, 25] that learn classification rules by minimizing the worst-case expected loss against an uncertainty set given by constraints on the expectation of a feature mapping $\Phi : \mathcal{X} \times \mathcal{Y} \to \mathbb{R}^m$ as

$$\mathcal{U} = \{\mathrm{p} \in \Delta(\mathcal{X} \times \mathcal{Y}) : |\mathbb{E}_\mathrm{p}\{\Phi(x, y)\} - \boldsymbol{\tau}| \preceq \boldsymbol{\lambda}\} \tag{1}$$

where $\boldsymbol{\tau}$ denotes a mean vector of expectation estimates and $\boldsymbol{\lambda}$ denotes a confidence vector. Feature mappings are vector-valued functions over $\mathcal{X} \times \mathcal{Y}$, e.g., one-hot encodings of labels [26] with instances represented by values from the last layers in a neural network [8, 27] or by random Fourier features (RFF) [28, 29].

Given the uncertainty set $\mathcal{U}$, MRC rules are solutions of the optimization problem

$$R(\mathcal{U}) = \min_{\mathrm{h} \in \mathrm{T}(\mathcal{X}, \mathcal{Y})} \max_{\mathrm{p} \in \mathcal{U}} \ell(\mathrm{h}, \mathrm{p}) \tag{2}$$

where $R(\mathcal{U})$ denotes the minimax risk and $\ell(\mathrm{h}, \mathrm{p})$ denotes the expected loss of classification rule $\mathrm{h}$ for distribution $\mathrm{p}$. In the following, we utilize the 0-1-loss so that $\ell(\mathrm{h}, \mathrm{p}) = \mathbb{E}_\mathrm{p}\{\mathbb{I}\{\mathrm{h}(x) \neq y\}\}$ and the expected loss with respect to the underlying distribution becomes the error probability of the classification rule. Deterministic MRCs assign each instance $x \in \mathcal{X}$ with the label $\mathrm{h}(x) \in \arg\max_{y \in \mathcal{Y}} \Phi(x, y)^\mathrm{T} \boldsymbol{\mu}^*$ where the parameter $\boldsymbol{\mu}^*$ is the solution of the convex optimization problem

$$\min_{\boldsymbol{\mu}} 1 - \boldsymbol{\tau}^\mathrm{T} \boldsymbol{\mu} + \max_{x \in \mathcal{X}, \mathcal{C} \subseteq \mathcal{Y}} \frac{\sum_{y \in \mathcal{C}} \Phi(x, y)^\mathrm{T} \boldsymbol{\mu} - 1}{|\mathcal{C}|} + \boldsymbol{\lambda}^\mathrm{T} |\boldsymbol{\mu}| \tag{3}$$

given by the Lagrange dual of (2) [24, 25].

The baseline approach of single-task learning obtains a classification rule $\mathrm{h}_j$ for each $j$-th task leveraging information only from the sample set $D_j = \{(x_{j,i}, y_{j,i})\}_{i=1}^{n_j}$ given by $n_j$ samples from distribution $\mathrm{p}_j$. In that case, IMRCs coincide with MRCs for standard supervised classification that obtain the mean and confidence vectors as

$$\boldsymbol{\tau}_j = \frac{1}{n_j} \sum_{i=1}^{n_j} \Phi(x_{j,i}, y_{j,i}), \quad \boldsymbol{\lambda}_j = \sqrt{\boldsymbol{s}_j}, \quad \boldsymbol{s}_j = \frac{\boldsymbol{\sigma}_j^2}{n_j} \tag{4}$$

with $\boldsymbol{\sigma}_j^2$ an estimate of $\mathbb{V}\mathrm{ar}_{\mathrm{p}_j}\{\Phi(x, y)\}$, e.g., the sample variance of the $n_j$ samples. The vector $\boldsymbol{s}_j$ describes the mean squared errors (MSEs) of the mean vector components and directly gives the confidence vector $\boldsymbol{\lambda}_j$ as shown in (4).

MRCs provide bounds for the minimax risk $R(\mathcal{U}_j)$ in terms of the smallest minimax risk as described in [24, 25, 30]. The smallest minimax risk is that corresponding to the ideal case of knowing mean vectors exactly, that is, the minimax risk corresponding with the uncertainty set $\mathcal{U}_j^\infty = \{\mathrm{p} \in \Delta(\mathcal{X} \times \mathcal{Y}) : \mathbb{E}_\mathrm{p}\{\Phi(x, y)\} = \boldsymbol{\tau}_j^\infty\}$ given by the expectation $\boldsymbol{\tau}_j^\infty = \mathbb{E}_{\mathrm{p}_j}\{\Phi(x, y)\}$. In the baseline approach of single-task learning, if $R_j^\infty$ and $\boldsymbol{\mu}_j^\infty$ denote the smallest minimax risk and the MRC parameter corresponding with $\mathcal{U}_j^\infty$, with probability at least $1 - \delta$ we have that

$$R(\mathcal{U}_j) \leq R_j^\infty + \frac{M(\kappa + 1)\sqrt{2\log(2m/\delta)}}{\sqrt{n_j}} \|\boldsymbol{\mu}_j^\infty\|_1 \tag{5}$$

where $M$ and $\kappa$ are such that $\|\Phi(x, y)\|_\infty \leq M$ for any $(x, y) \in \mathcal{X} \times \mathcal{Y}$ and $subG(\Phi_j) \preceq \kappa \boldsymbol{\sigma}_j$, $subG(\cdot)$ denotes the sub-Gaussian parameter of the argument components, and $\Phi_j$ denotes the random variable given by the feature mapping of samples from the $j$-th task. Inequality (5) is obtained using the bounds in [25] together with the Chernoff bound [31] for sub-Gaussian variables.

In the following sections, we describe techniques that obtain the mean and MSE vectors using forward and backward learning. Once such vectors are obtained, IMRC methods obtain the classifier parameter $\boldsymbol{\mu}_j$ for each $j$-th task solving the convex optimization problem in (3) that can be efficiently addressed using conventional methods [32, 33].

## 3 Forward learning with performance guarantees

This section presents the recursions that allow to obtain mean and MSE vectors for each task retaining information from preceding tasks. In addition, it characterizes the increase in ESS provided by forward learning in terms of the tasks' expected quadratic change and the number of tasks.

### 3.1 Forward learning

Let $\boldsymbol{\tau}_j^{\rightarrow}$ and $\boldsymbol{s}_j^{\rightarrow}$ denote the mean and MSE vectors for forward learning corresponding to the $j$-th task. The following recursions allow to obtain $\boldsymbol{\tau}_j^{\rightarrow}$ and $\boldsymbol{s}_j^{\rightarrow}$ for each $j$-th task using the vectors for the preceding task $\boldsymbol{\tau}_{j-1}^{\rightarrow}, \boldsymbol{s}_{j-1}^{\rightarrow}$ as

$$\boldsymbol{\tau}_j^{\rightarrow} = \boldsymbol{\tau}_j + \frac{\boldsymbol{s}_j}{\boldsymbol{s}_{j-1}^{\rightarrow} + \boldsymbol{s}_j + \boldsymbol{d}_j^2}\left(\boldsymbol{\tau}_{j-1}^{\rightarrow} - \boldsymbol{\tau}_j\right), \quad \boldsymbol{s}_j^{\rightarrow} = \left(\frac{1}{\boldsymbol{s}_j} + \frac{1}{\boldsymbol{s}_{j-1}^{\rightarrow} + \boldsymbol{d}_j^2}\right)^{-1} \tag{6}$$

with $\boldsymbol{\tau}_j$ and $\boldsymbol{s}_j$ given by (4) and $\boldsymbol{\tau}_1^{\rightarrow} = \boldsymbol{\tau}_1$ and $\boldsymbol{s}_1^{\rightarrow} = \boldsymbol{s}_1$.

The vector $\boldsymbol{d}_i^2$ assesses the expected quadratic change between consecutive tasks described by $\boldsymbol{w}_i = \boldsymbol{\tau}_i^\infty - \boldsymbol{\tau}_{i-1}^\infty$. Taking $\boldsymbol{d}_i^2 = \mathbb{E}\{\boldsymbol{w}_i^2\}$ and $\boldsymbol{\sigma}_i^2 = \mathbb{V}\mathrm{ar}_{\mathrm{p}_i}\{\Phi(x,y)\}$ for any $i$, the first recursion in (6) provides the unbiased linear estimator of the mean vector $\boldsymbol{\tau}_j^\infty$ based on $D_1, D_2, \ldots, D_j$ that has the minimum MSE, while the second recursion in (6) provides its MSE (see Appendix B for a detailed derivation). Vectors $\boldsymbol{\sigma}_i^2$ and $\boldsymbol{d}_i^2$ can be estimated online using the sample sets. In particular, $\boldsymbol{\sigma}_i^2$ can be estimated as the sample variance, while $\boldsymbol{d}_i^2$ can be estimated using sample averages as

$$\boldsymbol{d}_i^2 = \sum_{l=1}^{W} \frac{(\boldsymbol{\tau}_{i_l} - \boldsymbol{\tau}_{i_{l-1}})^2}{W} \text{ where } i_0, i_1, \ldots, i_W \text{ are the closest indexes to } i \in \{1, 2, \ldots, k\}. \quad (7)$$

Recursions in (6) obtain mean and MSE vectors for the $j$-th task by using information from preceding tasks and the $j$-th sample set $D_j$. Specifically, the first recursion in (6) obtains the mean vector $\boldsymbol{\tau}_j^\rightarrow$ by adding a correction to the sample average $\boldsymbol{\tau}_j$. This correction is proportional to the difference between $\boldsymbol{\tau}_j$ and $\boldsymbol{\tau}_{j-1}^\rightarrow$ with a proportionality constant that depends on the MSE vectors $\boldsymbol{s}_j, \boldsymbol{s}_{j-1}^\rightarrow$ and the expected quadratic change $\boldsymbol{d}_j^2$. In particular, if $\boldsymbol{s}_j \ll \boldsymbol{s}_{j-1}^\rightarrow + \boldsymbol{d}_j^2$, the mean vector is given by the sample average as in single-task learning, and if $\boldsymbol{s}_j \gg \boldsymbol{s}_{j-1}^\rightarrow + \boldsymbol{d}_j^2$, the mean vector is given by that of the preceding task. Note that for forward learning, at each step $k$, only the vectors for the last task $\boldsymbol{\tau}_k^\rightarrow$ and $\boldsymbol{s}_k^\rightarrow$ need to be obtained from those of the $(k-1)$-th task, the vectors for the remaining tasks stay the same as at step $k-1$ (see also Fig. 2 and Alg. 1 below).

### 3.2 Performance guarantees and effective sample sizes with forward learning

The following result provides bounds for the minimax risk for each task using forward learning.

**Theorem 1.** Let $\mathcal{U}_j^\rightarrow$ be the uncertainty set given by (1) using the mean and confidence vectors $\boldsymbol{\tau}_j^\rightarrow$ and $\boldsymbol{\lambda}_j^\rightarrow = \sqrt{\boldsymbol{s}_j^\rightarrow}$ provided by (6), and let $\kappa$ be such that $subG(\Phi_j) \preceq \kappa \boldsymbol{\sigma}_j$ and $subG(\boldsymbol{w}_j) \preceq \kappa \boldsymbol{d}_j$ for $j = 1, 2, \ldots, k$. Then, under the evolving task assumption of Section 2, we have with probability at least $1 - \delta$ that

$$R(\mathcal{U}_j^\rightarrow) \le R_j^\infty + \frac{M(\kappa+1)\sqrt{2\log(2m/\delta)}}{\sqrt{n_j^\rightarrow}} \left\| \boldsymbol{\mu}_j^\infty \right\|_1 \text{ for any } j \in \{1, 2, \ldots, k\} \quad (8)$$

with $n_1^\rightarrow = n_1$ and $n_j^\rightarrow \ge n_j + n_{j-1}^\rightarrow \frac{\|\boldsymbol{\sigma}_j^2\|_\infty}{\|\boldsymbol{\sigma}_j^2\|_\infty + n_{j-1}^\rightarrow \|\boldsymbol{d}_j^2\|_\infty}$ for $j \ge 2$.

*Proof.* See Appendix C. □

The value $n_j^\rightarrow$ in (8) is the ESS of the proposed IMRC method with forward learning since the bound in (8) coincides with that of single-task learning in (5) if the sample size for the $j$-th task is $n_j^\rightarrow$. The ESS of each task is obtained by adding a fraction of the ESS for the preceding task to the sample size. In particular, if $\boldsymbol{d}_j^2$ is large, the ESS is given by the sample size, while if $\boldsymbol{d}_j^2$ is small, the ESS is given by the sum of the sample size and the ESS of the preceding task.

The bound in (8) shows that recursions in (6) do not need to use very accurate values for $\boldsymbol{\sigma}_j$ and $\boldsymbol{d}_j$. Specifically, the coefficient $\kappa$ in (8) can be taken to be small as long as the values used for $\boldsymbol{\sigma}_j$ and $\boldsymbol{d}_j$ are not much lower than the sub-Gaussian parameters of $\Phi_j$ and $\boldsymbol{w}_j$, respectively. In particular, $\kappa$ is smaller than the maximum of $M/\min_{j,i}\{\sigma_j^{(i)}\}$ and $2M/\min_{j,i}\{d_j^{(i)}\}$ due to the bound for the sub-Gaussian parameter of bounded random variables (see e.g., Section 2.1.2 in [31]).

Theorem 1 shows the increase in ESS in terms of the ESS of the preceding task. The following result allows to directly quantify the ESS in terms of the tasks' expected quadratic change and the number of tasks.

**Theorem 2.** If $\|\boldsymbol{d}_j^2\|_\infty \le d^2$, $M \le 1$, and $n_j \ge n$ for $j = 1, 2, \ldots, k$, then for any $j \in \{1, 2, \ldots, k\}$, we have that the ESS in (8) can be taken so that it satisfies

$$n_j^\rightarrow \ge n \left(1 + \frac{(1+\alpha)^{2j-1} - 1 - \alpha}{\alpha(1+\alpha)^{2j-1} + \alpha}\right) \text{ with } \alpha = \frac{2}{\sqrt{1 + \frac{4}{nd^2}} - 1}. \quad (9)$$

In particular, if $nd^2 < 1/j^2$, we have that $n_j^\rightarrow \ge n\left(1 + \frac{j-1}{3}\right)$.

*Proof.* See Appendix D. □

The above theorem characterizes the increase in ESS provided by forward learning in terms of the tasks' expected quadratic change. Such increase grows monotonically with the number of preceding tasks $j$ as shown in (9) and becomes proportional to $j$ when the expected quadratic change is smaller than $1/(j^2 n)$. Figure 3 below further illustrates the increase in ESS with respect to the sample size $(n_j^{\rightarrow}/n)$ due forward learning in comparison with forward and backward learning.

## 4 Forward and backward learning with performance guarantees

This section presents the recursions that allow to obtain mean and MSE vectors for each task leveraging information from all the tasks. In addition, it characterizes the increase in ESS provided by forward and backward learning in terms of the tasks' expected quadratic change and the number of tasks.

Backward learning is more challenging than forward learning since the new task provides additional information for preceding tasks at each time step, while the information from preceding tasks is always the same. The techniques proposed below for backward learning effectively increase the ESS over time by carefully accounting for the new information at each step.

### 4.1 Forward and backward learning

At each step $k$, the proposed techniques learn to classify each $j$-th task leveraging information obtained from the $j$ preceding tasks (tasks $\{1, 2, \ldots, j\}$) and from the $k - j$ succeeding tasks (tasks $\{j+1, j+2, \ldots, k\}$). From preceding tasks, we obtain the forward mean and MSE vectors $\boldsymbol{\tau}_j^{\rightarrow}, \boldsymbol{s}_j^{\rightarrow}$ using recursions in (6), while from succeeding tasks, we obtain the backward mean and MSE vectors $\boldsymbol{\tau}_j^{\leftarrow k}, \boldsymbol{s}_j^{\leftarrow k}$ using recursions in (6) in retrodiction. Specifically, vectors $\boldsymbol{\tau}_j^{\leftarrow k}$ and $\boldsymbol{s}_j^{\leftarrow k}$ are obtained using the same recursion as for $\boldsymbol{\tau}_j^{\rightarrow}$ and $\boldsymbol{s}_j^{\rightarrow}$ in (6) with $\boldsymbol{s}_{j+1}^{\leftarrow k}, \boldsymbol{d}_{j+1}^2$, and $\boldsymbol{\tau}_{j+1}^{\leftarrow k}$ instead of $\boldsymbol{s}_{j-1}^{\rightarrow}, \boldsymbol{d}_j^2$, and $\boldsymbol{\tau}_{j-1}^{\rightarrow}$.

Let $\boldsymbol{\tau}_j^{\rightleftharpoons k}$ and $\boldsymbol{s}_j^{\rightleftharpoons k}$ denote the mean and MSE vectors for forward and backward learning corresponding to the $j$-th task for $j \in \{1, 2, \ldots, k\}$. The following recursions allow to obtain the mean and MSE vectors $\boldsymbol{\tau}_j^{\rightleftharpoons k}$ and $\boldsymbol{s}_j^{\rightleftharpoons k}$ for each $j$-th task using those vectors for forward learning $\boldsymbol{\tau}_j^{\rightarrow}, \boldsymbol{s}_j^{\rightarrow}$ and backward learning $\boldsymbol{\tau}_{j+1}^{\leftarrow k}, \boldsymbol{s}_{j+1}^{\leftarrow k}$ as

$$
\boldsymbol{\tau}_j^{\rightleftharpoons k} = \boldsymbol{\tau}_j^{\rightarrow} + \frac{\boldsymbol{s}_j^{\rightarrow}}{\boldsymbol{s}_j^{\rightarrow} + \boldsymbol{s}_{j+1}^{\leftarrow k} + \boldsymbol{d}_{j+1}^2} \left( \boldsymbol{\tau}_{j+1}^{\leftarrow k} - \boldsymbol{\tau}_j^{\rightarrow} \right), \quad \boldsymbol{s}_j^{\rightleftharpoons k} = \left( \frac{1}{\boldsymbol{s}_j^{\rightarrow}} + \frac{1}{\boldsymbol{s}_{j+1}^{\leftarrow k} + \boldsymbol{d}_{j+1}^2} \right)^{-1} \quad (10)
$$

with $\boldsymbol{\tau}_k^{\leftarrow k} = \boldsymbol{\tau}_k, \boldsymbol{s}_k^{\leftarrow k} = \boldsymbol{s}_k$ and $\boldsymbol{\tau}_k^{\rightleftharpoons k} = \boldsymbol{\tau}_k^{\rightarrow}, \boldsymbol{s}_k^{\rightleftharpoons k} = \boldsymbol{s}_k^{\rightarrow}$. Analogously to the case of forward learning in Section 3.1, taking $\boldsymbol{d}_i^2 = \mathbb{E}\{\boldsymbol{w}_i^2\}$ and $\boldsymbol{\sigma}_i^2 = \mathbb{V}\text{ar}_{\text{p}_i}\{\Phi(x, y)\}$ for any $i$, the first recursion in (10) provides the unbiased linear estimator of the mean vector $\boldsymbol{\tau}_j^{\infty}$ based on $D_1, D_2, \ldots, D_j$ and $D_{j+1}, D_{j+2}, \ldots, D_k$ that has the minimum MSE, while the second recursion in (10) provides its MSE (see Appendix B for a detailed derivation).

Recursions in (10) obtain at step $k$ the mean and MSE vectors for the $j$-th task by retaining information from preceding tasks and acquiring information from the new task. Specifically, the first recursion in (10) obtains the mean vector $\boldsymbol{\tau}_j^{\rightleftharpoons k}$ by adding a correction to the mean vector of the corresponding task $\boldsymbol{\tau}_j^{\rightarrow}$ obtained for forward learning. This correction is proportional to the difference between $\boldsymbol{\tau}_j^{\rightarrow}$ and $\boldsymbol{\tau}_{j+1}^{\leftarrow k}$ with a proportionality constant that depends on the MSE vectors $\boldsymbol{s}_j^{\rightarrow}, \boldsymbol{s}_{j+1}^{\leftarrow k}$ and the expected quadratic change $\boldsymbol{d}_{j+1}^2$. In particular, if $\boldsymbol{s}_j^{\rightarrow} \ll \boldsymbol{s}_{j+1}^{\leftarrow k} + \boldsymbol{d}_{j+1}^2$, the mean vector is given by that of the corresponding task for forward learning, and if $\boldsymbol{s}_j^{\rightarrow} \gg \boldsymbol{s}_{j+1}^{\leftarrow k} + \boldsymbol{d}_{j+1}^2$, the mean vector is given by that of the next task for backward learning.

### 4.2 Implementation

This section describes the implementation of the proposed IMRCs with forward and backward learning and its computational and memory complexities.

Figure 2 depicts the flow diagram for the proposed IMRC methodology. The presented techniques carefully avoid the repeated usage of the same information from the sequence

of tasks. At each step $k$, the IMRC method obtains forward mean vector $\boldsymbol{\tau}_k^{\rightarrow}$ for the $k$-th task leveraging information from preceding tasks using the forward mean vector $\boldsymbol{\tau}_{k-1}^{\rightarrow}$ and the sample average $\boldsymbol{\tau}_k$. Reciprocally, backward mean vectors $\boldsymbol{\tau}_{j+1}^{\leftharpoonup k}$ for each $j$-th task are obtained leveraging information from the $k$-th task through the sample average $\boldsymbol{\tau}_k$.

Then, the forward and backward mean vectors $\boldsymbol{\tau}_j^{\rightleftharpoons k}$ are obtained from the forward mean vectors $\boldsymbol{\tau}_j^{\rightarrow}$ and the backward mean vectors $\boldsymbol{\tau}_{j+1}^{\leftharpoonup k}$. In particular, $\boldsymbol{\tau}_j^{\rightarrow}$ provides the information from the preceding tasks $1, 2, \ldots, j$, while $\boldsymbol{\tau}_{j+1}^{\leftharpoonup k}$ provides the information from the succeeding tasks $j + 1, j + 2, \ldots, k$. At each step $k$, the IMRC method obtains forward and backward mean vectors $\boldsymbol{\tau}_j^{\rightleftharpoons k}$ for $j = k - b, k - b + 1, \ldots, k$ with $b$ the number of backward steps. In particular, if $b = 0$, IMRC carries out only forward learning. Note that, at each step $k$, the proposed IMRC methods only need to retain the forward mean vectors $\boldsymbol{\tau}_j^{\rightarrow}$ and sample averages $\boldsymbol{\tau}_j$ for $j = k - b, k - b + 1, \ldots, k$.

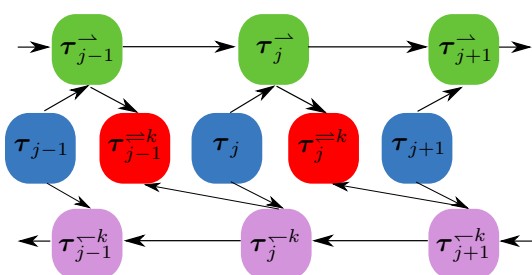

Figure 2: Diagram for IMRC methodology.

---

**Algorithm 1** IMRC at step $k$

---

**Input:** $D_k$ from new task, and $\boldsymbol{\tau}_j, \boldsymbol{s}_j, \boldsymbol{\tau}_j^{\rightarrow}, \boldsymbol{s}_j^{\rightarrow}$ for $k - b \leq j \leq k - 1$ from previous $b - 1$ steps
**Output:** $\boldsymbol{\mu}_j$ for $k - b \leq j \leq k, \boldsymbol{\tau}_k, \boldsymbol{s}_k, \boldsymbol{\tau}_k^{\rightarrow}, \boldsymbol{s}_k^{\rightarrow}$
  Obtain sample average and MSE vectors $\boldsymbol{\tau}_k^{\leftharpoonup k} = \boldsymbol{\tau}_k, \boldsymbol{s}_k^{\leftharpoonup k} = \boldsymbol{s}_k$ using the sample set $D_k$       ▷ Single-task
  Estimate the tasks' expected quadratic change $\boldsymbol{d}_k^2$ using (7)
  Obtain the forward mean and MSE vectors $\boldsymbol{\tau}_k^{\rightarrow}, \boldsymbol{s}_k^{\rightarrow}$ using (6)       ▷ Forward
  Take $\boldsymbol{\lambda}_k^{\rightarrow} = \sqrt{\boldsymbol{s}_k^{\rightarrow}}$ and obtain classifier parameter $\boldsymbol{\mu}_k$ solving the optimization problem (3)
  **for** $j = k - 1, k - 2, \ldots, k - b$ **do**
    Estimate the tasks' expected quadratic change $\boldsymbol{d}_j^2$ using (7)
    Obtain backward mean and MSE vectors $\boldsymbol{\tau}_{j+1}^{\leftharpoonup k}, \boldsymbol{s}_{j+1}^{\leftharpoonup k}$ using (6) in retrodiction       ▷ Backward
    Obtain mean and MSE vectors $\boldsymbol{\tau}_j^{\rightleftharpoons k}, \boldsymbol{s}_j^{\rightleftharpoons k}$ using (10)       ▷ Forward and backward
    Take $\boldsymbol{\lambda}_j^{\rightleftharpoons k} = \sqrt{\boldsymbol{s}_j^{\rightleftharpoons k}}$ and obtain classifier parameters $\boldsymbol{\mu}_j$ solving the optimization problem (3)

---

Algorithm 1 details the implementation of the proposed IMRCs at each step. For $k$ steps, IMRCs have computational complexity $\mathcal{O}((b+1)Kmk)$ and memory complexity $\mathcal{O}((b+k)m)$ where $K$ is the number of iterations used for the convex optimization problem (3), $m$ is the length of the feature vector, and $b$ is the number of backward steps. The complexity of forward and backward learning increases proportionally to the number of backward steps that can be taken to be rather small, as shown in the following. Even more efficient implementations can be obtained using Rauch-Tung-Striebel recursions (see e.g., Section 7.2 in [34]) that can be used to obtain $\boldsymbol{\tau}_j^{\rightleftharpoons k}$ from $\boldsymbol{\tau}_{j+1}^{\rightleftharpoons k}$, as shown in Appendix E.

### 4.3 Performance guarantees and effective sample sizes with forward and backward learning

The following result provides bounds for the minimax risk for each task using forward and backward learning

**Theorem 3.** Let $\mathcal{U}_j^{\rightleftharpoons k}$ be the uncertainty set given by (1) using the mean and confidence vector $\boldsymbol{\tau}_j^{\rightleftharpoons k}$ and $\boldsymbol{\lambda}_j^{\rightleftharpoons k} = \sqrt{\boldsymbol{s}_j^{\rightleftharpoons k}}$ provided by (10), and let $\kappa$ and $n_j^{\rightarrow}$ be as in Theorem 1. Then, under the evolving task assumption of Section 2, we have with probability at least $1 - \delta$ that

$$R(\mathcal{U}_j^{\rightleftharpoons k}) \leq R_j^{\infty} + \frac{M(\kappa + 1)\sqrt{2\log(2m/\delta)}}{\sqrt{n_j^{\rightleftharpoons k}}}\left\|\boldsymbol{\mu}_j^{\infty}\right\|_1 \text{ for any } j \in \{1, 2, \ldots, k\} \qquad (11)$$

with $n_k^{\rightleftharpoons k} = n_k^{\rightarrow}$ and $n_j^{\rightleftharpoons k} \geq n_j^{\rightarrow} + n_{j+1}^{\rightleftharpoons k} \frac{\|\boldsymbol{\sigma}_j^2\|_\infty}{\|\boldsymbol{\sigma}_j^2\|_\infty + n_{j+1}^{\rightleftharpoons k}\|\boldsymbol{d}_{j+1}^2\|_\infty}$ for $j \leq k-1$, where the backward

ESSs satisfy $n_k^{\rightleftharpoons k} = n_k$ and $n_j^{\rightleftharpoons k} \geq n_j + n_{j+1}^{\rightleftharpoons k} \frac{\|\boldsymbol{\sigma}_j^2\|_\infty}{\|\boldsymbol{\sigma}_j^2\|_\infty + n_{j+1}^{\rightleftharpoons k}\|\boldsymbol{d}_{j+1}^2\|_\infty}$.

*Proof.* See Appendix F. $\qquad\square$

Theorem 3 shows that the methods proposed can increase ESS of each task by leveraging information from all the tasks. In particular, the bounds for forward and backward learning provided by inequality (11) are significantly lower than those for forward learning in Theorem 1. The ESS of each task is obtained by adding a fraction of the ESS for the next task to the ESS of the corresponding task using forward learning. In particular, if $\boldsymbol{d}_j^2$ is large, the ESS is given by that with forward learning, while if $\boldsymbol{d}_j^2$ is small, the ESS is given by the sum of the ESS using forward learning and the ESS of the next task.

Theorem 3 shows the increase in ESS in terms of the ESS with forward learning and the ESS of the next task. The following result allows to directly quantify the ESS in terms of the tasks' expected quadratic change and the number of tasks.

**Theorem 4.** *If* $\|\boldsymbol{d}_j^2\|_\infty \leq d^2$, $M \leq 1$, *and* $n_j \geq n$ *for* $j = 1, 2, \ldots, k$, *then for any* $j \in \{1, 2, \ldots, k\}$, *we have that the ESS in* (11) *can be taken so that it satisfies*

$$n_j^{\rightleftharpoons k} \geq n\left(1 + \frac{(1+\alpha)^{2j-1} - 1 - \alpha}{\alpha(1+\alpha)^{2j-1} + \alpha} + \frac{(1+\alpha)^{2(k-j)+1} - 1 - \alpha}{\alpha(1+\alpha)^{2(k-j)+1} + \alpha}\right) \text{ with } \alpha = \frac{2}{\sqrt{1 + \frac{4}{nd^2}} - 1}.$$
(12)

*In particular, for* $j \geq 2$, *we have that*

$$n_j^{\rightleftharpoons k} \geq n_j^{\rightarrow} + n\frac{j(k-j)}{j + 2(k-j)} \geq n\left(1 + \frac{j-1}{3} + \frac{j(k-j)}{j + 2(k-j)}\right) \quad \text{if} \quad nd^2 < \frac{1}{j^2}$$

$$n_j^{\rightleftharpoons k} \geq n_j^{\rightarrow} + \frac{1}{5}\sqrt{\frac{n}{d^2}} \qquad \geq n\left(1 + \frac{2}{5\sqrt{nd^2}}\right) \qquad \text{if} \quad \frac{1}{j^2} \leq nd^2 < 1$$

$$n_j^{\rightleftharpoons k} \geq n_j^{\rightarrow} + \frac{1}{3d^2} \qquad \geq n\left(1 + \frac{2}{3nd^2}\right) \qquad \text{if} \quad nd^2 \geq 1.$$

*Proof.* See Appendix G. $\qquad\square$

The above theorem characterizes the increase in ESS provided by forward and backward learning. This increase grows monotonically with the number of preceding tasks $j$ and with the number of succeeding tasks $k - j$ as shown in (12). In addition, it becomes proportional to the total number of tasks $k$ when the expected quadratic change is smaller than $1/(j^2 n)$ and $j \geq k/2$. Figure 3 further illustrates the increase in ESS with respect to the sample size $(n_j^{\rightleftharpoons k}/n)$ due to forward and backward learning in comparison with forward learning. As the figure shows, the increase in ESS can be classified into three regimes depending on the sample size $n$ and the expected quadratic change $d^2$. The ESS is only marginaly larger than the sample size for sizeble values of $nd^2$ (large samples sizes or drastic changes in the distribution); the ESS quickly increases when $nd^2$ becomes small (reduced sample sizes and moderate changes in the distribution); and the ESS becomes proportional to the total number of tasks $k$ if $nd^2$ is rather small (very small sample sizes and very slow changes in the distribution).

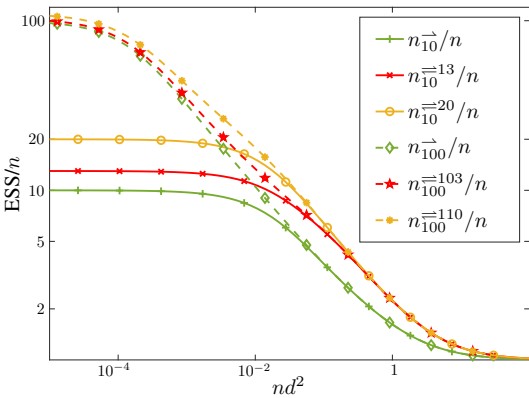

Figure 3: ESS increase provided by forward and backward learning.

Table 1: Classification error, standard deviation, and running time of the proposed IMRC method in comparison with existing techniques using $n = 10$ samples per task.

| Method | GEM [5] | MER [17] | ELLA [4] | EWC [6] | Condor [9] | DriftSurf [10] | AUE [16] | IMRC |
|---|---|---|---|---|---|---|---|---|
| Yearbook [1] | $.18 \pm .03$ | $.16 \pm .03$ | $.45 \pm .01$ | $.47 \pm .05$ | $.14 \pm .01$ | $.33 \pm .02$ | $.33 \pm .02$ | $\mathbf{.13} \pm .04$ |
| I. noise [35] | $.39 \pm .08$ | $.17 \pm .03$ | $.48 \pm .05$ | $.47 \pm .04$ | $.16 \pm .02$ | $.48 \pm .02$ | $.48 \pm .02$ | $\mathbf{.15} \pm .03$ |
| DomainNet [36] | $.69 \pm .05$ | $.38 \pm .04$ | $.67 \pm .05$ | $.75 \pm .04$ | $.45 \pm .04$ | $\mathbf{.32} \pm .02$ | $.33 \pm .02$ | $.34 \pm .06$ |
| UTKFaces [37] | $.12 \pm .00$ | $.17 \pm .09$ | $.19 \pm .12$ | $.12 \pm .00$ | $.23 \pm .00$ | $.12 \pm .00$ | $.12 \pm .00$ | $\mathbf{.10} \pm .01$ |
| R. MNIST [38] | $\mathbf{.36} \pm .06$ | $.37 \pm .09$ | $.48 \pm .05$ | $.48 \pm .01$ | $.45 \pm .02$ | $.48 \pm .02$ | $.48 \pm .02$ | $\mathbf{.36} \pm .01$ |
| CLEAR [39] | $.57 \pm .10$ | $.10 \pm .03$ | $.61 \pm .06$ | $.65 \pm .03$ | $.35 \pm .00$ | $.33 \pm .00$ | $.33 \pm .01$ | $\mathbf{.09} \pm .03$ |
| P. Supply [10] | $.46 \pm .01$ | $.47 \pm .01$ | $.41 \pm .07$ | $.47 \pm .00$ | $.36 \pm .01$ | $.45 \pm .02$ | $.45 \pm .02$ | $\mathbf{.30} \pm .01$ |
| Usenet1 [40] | $.48 \pm .02$ | $.48 \pm .02$ | $.39 \pm .05$ | $.48 \pm .02$ | $.48 \pm .02$ | $.40 \pm .01$ | $.40 \pm .01$ | $\mathbf{.32} \pm .03$ |
| Usenet2 [40] | $.36 \pm .01$ | $\mathbf{.33} \pm .01$ | $.39 \pm .03$ | $.48 \pm .02$ | $.42 \pm .01$ | $.35 \pm .03$ | $.39 \pm .03$ | $\mathbf{.33} \pm .02$ |
| German [41] | $.38 \pm .16$ | $.38 \pm .10$ | $\mathbf{.33} \pm .04$ | $.35 \pm .11$ | $.41 \pm .02$ | $.36 \pm .00$ | $.35 \pm .02$ | $.34 \pm .01$ |
| Spam [2] | $.25 \pm .05$ | $\mathbf{.13} \pm .02$ | $.28 \pm .07$ | $.33 \pm .11$ | $.22 \pm .02$ | $.24 \pm .03$ | $.25 \pm .03$ | $\mathbf{.13} \pm .01$ |
| Covert. [10] | $.09 \pm .01$ | $\mathbf{.08} \pm .00$ | $.13 \pm .01$ | $.09 \pm .00$ | $.09 \pm .00$ | $.10 \pm .00$ | $.11 \pm .00$ | $\mathbf{.08} \pm .00$ |
| Ave. running time | 0.238 | 0.250 | 0.055 | 0.313 | 0.184 | 0.128 | 0.044 | 0.275 |

## 5 Numerical results

This section first compares the classification performance of IMRCs with existing techniques using multiple datasets, then we show the performance improvement of the presented IMRCs due to forward and backward learning. In Appendix H, we provide additional implementation details and numerical results. The implementation of the proposed IMRCs is available on the web `https://github.com/MachineLearningBCAM/IMRCs-for-incremental-learning-NeurIPS-2023`.

The proposed method is evaluated using 12 public datasets composed by evolving tasks [10, 1, 35–41, 2]. Six datasets are formed by images with characteristics/quality/realism that change over time; while the rest are formed by tabular data that are segmented by time (see further details in Appendix H). For instance, each task in the "Yearbook" dataset corresponds to portrait's photographs from one year from 1905 to 2013 and the goal is to predict males and females; and each task in the "CLEAR" dataset corresponds to images with a natural temporal evolution of visual concepts per year from 2004 to 2014 and the goal is to predict if an image is soccer, hockey, or racing. For the image datasets, instances are represented by pixel values in "Rotated MNIST" dataset, and by the last layer of the ResNet18 pre-trained network [42] in the remaining datasets; while for the tabular datasets, instances are represented by 200 RFFs [28, 29] with scaling parameter $\sigma^2 = 10$.

The proposed IMRC method is compared with 7 state-of-the-art-techniques: 2 techniques of continual learning based on experience replay [5, 17], a technique of continual learning based on regularization [6], a technique of continual learning based on dynamic architectures [4], 2 techniques of concept drift adaptation based on weight factors [9, 16], and a technique of concept drift adaptation based on sliding windows [10]. The hyper-parameters in all methods are set to the default values provided by the authors. IMRCs are implemented using $b = 3$ backward steps and the expected quadratic change $\boldsymbol{d}_j^2$ is estimated using $W = 2$ in (7). In Appendix H, among other additional results, we study the change in classification error and processing time achieved by varying the number $b$ of backward steps.

In the first set of numerical results, we compare the performance of the proposed IMRCs with existing techniques using $n = 10$ samples per task. These numerical results are obtained computing the average classification error over all the time steps and tasks in 50 random instantiations of data samples. As can be observed in Table 1, IMRCs can significantly improve performance in evolving tasks with respect to existing methods. Certain techniques of continual learning or concept drift adaptation achieve top performance in specific datasets, but the performance improvement of the methods presented are realized over general datasets with evolving tasks. The results for $n = 10$ samples per task in Table 1 are complemented with results for $n = 50, 100,$ and $150$ samples per task in Tables 4, 5, and 6 in Appendix H. These results show that the improved performance of IMRCs compared to the state-of-the-art techniques is similar for different sample sizes. In addition, Table 1 shows that the average running time per task of IMRC methods is similar to other state-of-the-art methods.

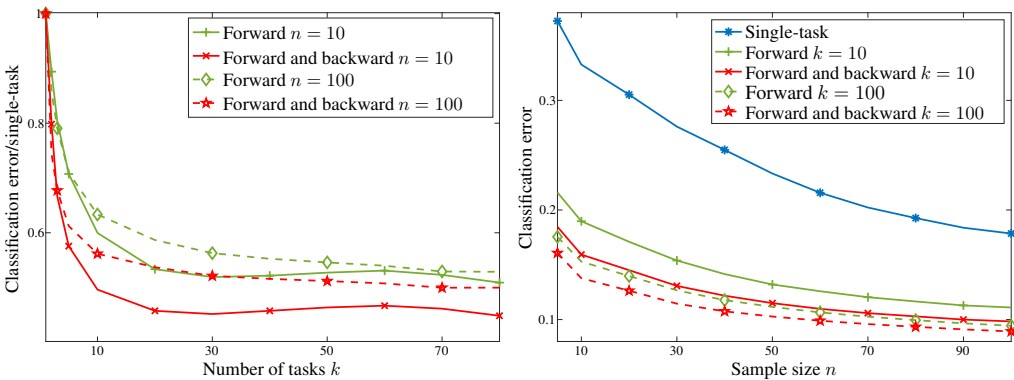

(a) Classification error per number of tasks in "Year- (b) Classification error per sample size in "Year-
book" dataset.                                          book" dataset.

Figure 4: Forward and backward learning can sharply boost performance and ESS as tasks arrive.

In the second set of numerical results, we analyze the contribution of forward and backward learn-
ing to the final performance of IMRCs. In particular, we show the relationship among classification
error, number of tasks, and sample size for single-task, forward, and forward and backward learn-
ing. These numerical results are obtained averaging, for each number of tasks and sample size, the
classification errors achieved with 10 random instantiations of data samples in "Yearbook" dataset
(see Appendix H for further details). Figure 4a shows the classification error of IMRC method di-
vided by the classification error of single-task learning for different number of tasks with $n = 10$
and $n = 100$ sample sizes. Such figure shows that forward and backward learning can significantly
improve the performance of IMRCs as tasks arrive. In addition, Figure 4b shows the classification
error of IMRC method for different sample sizes with $k = 10$ and $k = 100$ tasks. Such figure shows
that IMRCs with forward and backward learning for $k = 100$ tasks and $n = 10$ samples achieve
significantly better results than single-task learning using $n = 100$ samples. In particular, the ex-
periments show that the methods proposed can effectively exploit backward learning that results in
enhanced classification error in all the experimental results.

## 6    Conclusion

The paper proposes IMRCs that effectively exploit forward and backward learning and account for
evolving tasks by using a sequence of uncertainty sets that can contain the true underlying distribu-
tions. In addition, the paper analytically characterizes the increase in ESS achieved by the proposed
techniques in terms of the tasks' expected quadratic change and number of tasks. The numerical
results assess the performance improvement of IMRCs with respect to existing methods using multi-
ple datasets, sample sizes, and number of tasks. The proposed methodology for incremental learning
with evolving tasks can lead to techniques that further approach the humans' ability to learn from
few examples and to continuously improve on tasks that arrive over time.

## Acknowledgments

Funding in direct support of this work has been provided by projects PID2022-137063NB-
I00, PID2022-137442NB-I00, CNS2022-135203, and CEX2021-001142-S funded by
MCIN/AEI/10.13039/501100011033 and the European Union "NextGenerationEU"/PRTR,
BCAM Severo Ochoa accreditation CEX2021-001142-S / MICIN / AEI/ 10.13039/501100011033
funded by the Ministry of Science and Innovation, and programes ELKARTEK, IT1504-22, and
BERC-2022-2025 funded by the Basque Government.

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

# A Main notations used in the paper.

In this section we summarize the notations used in the paper. Table 2 shows the notation used for the mean vector, the confidence vector, the MSE vector, the ESS, the uncertainty set, and the minimax risk using single task, forward, backward, and forward and backward learning.

Table 2: Main notations used in the paper

|  | Single task | Forward | Backward | Forward and Backward |
|---|---|---|---|---|
| Mean vector | $\boldsymbol{\tau}_j$ | $\boldsymbol{\tau}_j^{\rightarrow}$ | $\boldsymbol{\tau}_j^{\leftarrow k}$ | $\boldsymbol{\tau}_j^{\rightleftarrows k}$ |
| Confidence vector | $\boldsymbol{\lambda}_j$ | $\boldsymbol{\lambda}_j^{\rightarrow}$ | $\boldsymbol{\lambda}_j^{\leftarrow k}$ | $\boldsymbol{\lambda}_j^{\rightleftarrows k}$ |
| MSE vector | $\boldsymbol{s}_j$ | $\boldsymbol{s}_j^{\rightarrow}$ | $\boldsymbol{s}_j^{\leftarrow k}$ | $\boldsymbol{s}_j^{\rightleftarrows k}$ |
| ESS | $\boldsymbol{n}_j$ | $\boldsymbol{n}_j^{\rightarrow}$ | $\boldsymbol{n}_j^{\leftarrow k}$ | $\boldsymbol{n}_j^{\rightleftarrows k}$ |
| Uncertainty set | $\mathcal{U}_j$ | $\mathcal{U}_j^{\rightarrow}$ | $\mathcal{U}_j^{\leftarrow k}$ | $\mathcal{U}_j^{\rightleftarrows k}$ |
| Minimax risk | $R(\mathcal{U}_j)$ | $R(\mathcal{U}_j^{\rightarrow})$ | $R(\mathcal{U}_j^{\leftarrow k})$ | $R(\mathcal{U}_j^{\rightleftarrows k})$ |

# B Derivation of recursions in (6) for forward learning and recursions in (10) for forward and backward learning

This section shows how recursions in (6) and recursions in (10) are obtained using those for filtering and smoothing in linear dynamical systems.

The mean vectors evolve over time steps through the linear dynamical system

$$\boldsymbol{\tau}_j^{\infty} = \boldsymbol{\tau}_{j-1}^{\infty} + \boldsymbol{w}_j \tag{13}$$

where vectors $\boldsymbol{w}_j$ for $j \in \{2, 3, \ldots, k\}$ are independent and zero-mean because $\mathrm{p}_j - \mathrm{p}_{j-1}$ are independent and zero-mean. In addition, each state variable $\boldsymbol{\tau}_j^{\infty}$ is observed at each step $j$ through $\boldsymbol{\tau}_j$ that is the sample average of i.i.d. samples from $\mathrm{p}_j$, so that we have

$$\boldsymbol{\tau}_j = \boldsymbol{\tau}_j^{\infty} + \boldsymbol{v}_j \tag{14}$$

where vectors $\boldsymbol{v}_j$ for $j \in \{1, 2, \ldots, k\}$ are independent and zero mean, and independent of $\boldsymbol{w}_j$ for $j \in \{1, 2, \ldots, k\}$. Therefore, equations (13) and (14) above describe a linear dynamical system (state-space model with white noise processes) [34]. For such systems, the Kalman filter recursions provide the unbiased linear estimator with minimum MSE based on samples corresponding to preceding steps $D_1, D_2, \ldots, D_j$, and fixed-lag smoother recursions provide the unbiased linear estimator with minimum MSE based on samples corresponding to preceding and succeeding steps $D_1, D_2, \ldots, D_k$ [34]. Then, equations in (6) and equations in (10) are obtained after some algebra from the Kalman filter recursions and fixed-lag smoother recursions, respectively.

# C Proof of Theorem 1

*Proof.* To obtain bound in (8) we first prove that the mean vector estimate and the MSE vector given by (6) satisfy

$$\mathbb{P}\left\{|\tau_j^{\infty(i)} - \tau_j^{\rightarrow(i)}| \leq \kappa \sqrt{2 s_j^{\rightarrow(i)} \log\left(\frac{2m}{\delta}\right)}\right\} \geq (1 - \delta) \tag{15}$$

for any component $i = 1, 2, \ldots, m$. Then, we prove that $\|\sqrt{\boldsymbol{s}_j^{\rightarrow}}\|_\infty \leq M/\sqrt{n_j^{\rightarrow}}$ for $j \in \{1, 2, \ldots, k\}$, where the ESSs satisfy $n_1^{\rightarrow} = n_1$ and $n_j^{\rightarrow} \geq n_j + n_{j-1}^{\rightarrow} \frac{\|\boldsymbol{\sigma}_j^2\|_\infty}{\|\boldsymbol{\sigma}_j^2\|_\infty + n_{j-1}^{\rightarrow}\|\boldsymbol{d}_j^2\|_\infty}$ for $j \geq 2$.

To obtain inequality (15), we prove by induction that each component $i = 1, 2, \ldots, m$ of the error in the mean vector estimate $z_j^{\rightarrow(i)} = \tau_j^{\infty(i)} - \tau_j^{\rightarrow(i)}$ is sub-Gaussian with parameter $\eta_j^{\rightarrow(i)} \leq \kappa \sqrt{s_j^{\rightarrow(i)}}$. Firstly, for $j = 1$, we have that

$$z_1^{\rightarrow(i)} = \tau_1^{\infty(i)} - \tau_1^{\rightarrow(i)} = \tau_1^{\infty(i)} - \tau_1^{(i)}.$$

Since the bounded random variable $\Phi_1^{(i)}$ is sub-Gaussian with parameter $\sigma(\Phi_1^{(i)})$, then the error in the mean vector estimate $z_1^{\rightarrow(i)}$ is sub-Gaussian with parameter that satisfies

$$\left(\eta_1^{\rightarrow(i)}\right)^2 = \frac{\sigma\left(\Phi_1^{(i)}\right)^2}{n_1} \leq \frac{\kappa^2 \sigma_1^{2(i)}}{n_1} = \kappa^2 s_1^{(i)}.$$

If $z_{j-1}^{\rightarrow(i)} = \tau_{j-1}^{\infty}{}^{(i)} - \tau_{j-1}^{\rightarrow(i)}$ is sub-Gaussian with parameter $\eta_{j-1}^{\rightarrow(i)} \leq \kappa\sqrt{s_{j-1}^{\rightarrow(i)}}$ for any $i = 1, 2, \ldots, m$, then using the recursions in (6) we have that

$$z_j^{\rightarrow(i)} = \tau_j^{\infty(i)} - \tau_j^{\rightarrow(i)} = \tau_{j-1}^{\infty}{}^{(i)} + w_j^{(i)} - \tau_j^{(i)} - \frac{s_j^{(i)}}{s_{j-1}^{\rightarrow(i)} + s_j^{(i)} + d_j^{2(i)}}\left(\tau_{j-1}^{\rightarrow(i)} - \tau_j^{(i)}\right)$$

$$= \tau_{j-1}^{\infty}{}^{(i)} + w_j^{(i)} - \tau_{j-1}^{\rightarrow(i)} + \left(1 - \frac{s_j^{(i)}}{s_{j-1}^{\rightarrow(i)} + s_j^{(i)} + d_j^{2(i)}}\right)\left(\tau_{j-1}^{\rightarrow(i)} - \tau_j^{(i)}\right)$$

$$= \tau_{j-1}^{\infty}{}^{(i)} + w_j^{(i)} - \tau_{j-1}^{\rightarrow(i)} - \frac{s_j^{\rightarrow(i)}}{s_j^{(i)}}\left(\tau_j^{(i)} - \tau_{j-1}^{\rightarrow(i)}\right)$$

since $\boldsymbol{w}_j = \boldsymbol{\tau}_j^{\infty} - \boldsymbol{\tau}_{j-1}^{\infty}$. If $\boldsymbol{v}_j = \boldsymbol{\tau}_j - \boldsymbol{\tau}_j^{\infty}$, the error in the mean vector estimate is given by

$$z_j^{\rightarrow(i)} = \tau_{j-1}^{\infty}{}^{(i)} + w_j^{(i)} - \tau_{j-1}^{\rightarrow(i)} - \frac{s_j^{\rightarrow(i)}}{s_j^{(i)}}\left(\tau_j^{\infty(i)} + v_j^{(i)} - \tau_{j-1}^{\rightarrow(i)}\right)$$

$$= \tau_{j-1}^{\infty}{}^{(i)} + w_j^{(i)} - \tau_{j-1}^{\rightarrow(i)} - \frac{s_j^{\rightarrow(i)}}{s_j^{(i)}}\left(\tau_{j-1}^{\infty}{}^{(i)} + w_j^{(i)} + v_j^{(i)} - \tau_{j-1}^{\rightarrow(i)}\right)$$

$$= \left(1 - \frac{s_j^{\rightarrow(i)}}{s_j^{(i)}}\right)z_{j-1}^{\rightarrow(i)} + \left(1 - \frac{s_j^{\rightarrow(i)}}{s_j^{(i)}}\right)\left(w_j^{(i)}\right) - \frac{s_j^{\rightarrow(i)}}{s_j^{(i)}}v_j^{(i)}$$

where $w_j^{(i)}$ and $v_j^{(i)}$ are sub-Gaussian with parameter $\sigma(w_j^{(i)})$ and $\sigma\left(\Phi_j^{(i)}\right)/\sqrt{n_j}$, respectively. Therefore, we have that $z_j^{\rightarrow(i)}$ is sub-Gaussian with parameter $\eta_j^{\rightarrow(i)}$ that satisfies

$$\left(\eta_j^{\rightarrow(i)}\right)^2 = \left(1 - \frac{s_j^{\rightarrow(i)}}{s_j^{(i)}}\right)^2\left(\eta_{j-1}^{\rightarrow(i)}\right)^2 + \left(1 - \frac{s_j^{\rightarrow(i)}}{s_j^{(i)}}\right)^2\sigma\left(w_j^{(i)}\right)^2 + \left(\frac{s_j^{\rightarrow(i)}}{s_j^{(i)}}\right)^2\frac{\sigma\left(\Phi_j^{(i)}\right)^2}{n_j}$$

since $\boldsymbol{z}_{j-1}^{\rightarrow}$, $\boldsymbol{w}_j$, and $\boldsymbol{v}_j$ are independent. Using that $\eta_{j-1}^{\rightarrow(i)} \leq \kappa\sqrt{s_{j-1}^{\rightarrow(i)}}$ and the definition of $\kappa$, we have that

$$\left(\eta_j^{\rightarrow(i)}\right)^2 \leq \left(1 - \frac{s_j^{\rightarrow(i)}}{s_j^{(i)}}\right)^2\kappa^2 s_{j-1}^{\rightarrow(i)} + \left(1 - \frac{s_j^{\rightarrow(i)}}{s_j^{(i)}}\right)^2\kappa^2 d_j^{2(i)} + \left(\frac{s_j^{\rightarrow(i)}}{s_j^{(i)}}\right)^2\kappa^2\frac{\sigma_j^{2(i)}}{n_j}$$

$$\leq \left(1 - \frac{s_j^{\rightarrow(i)}}{s_j^{(i)}}\right)^2\kappa^2\left(\left(\frac{1}{s_j^{\rightarrow(i)}} - \frac{1}{s_j^{(i)}}\right)^{-1} + d_j^{2(i)}\right) + \frac{\left(s_j^{\rightarrow(i)}\right)^2}{s_j^{(i)}}\kappa^2 \qquad (16)$$

$$= \left(1 - \frac{s_j^{\rightarrow(i)}}{s_j^{(i)}}\right)\kappa^2 s_j^{\rightarrow(i)} + \kappa^2\frac{\left(s_j^{\rightarrow(i)}\right)^2}{s_j^{(i)}}$$

where (16) is obtained using the second recursion in (6).

The inequality in (15) is obtained using the union bound together with the Chernoff bound (concentration inequality) [31] for the random variables $z_j^{\rightarrow(i)}$ that are sub-Gaussian with parameter $\eta_j^{\rightarrow(i)}$.

Now, we prove by induction that, for any $j$, $\|\sqrt{\boldsymbol{s}_j^{\rightarrow}}\|_\infty \leq M/\sqrt{n_j^{\rightarrow}}$ where the ESSs satisfy $n_1^{\rightarrow} = n_1$ and $n_j^{\rightarrow} \geq n_j + n_{j-1}^{\rightarrow}\frac{\|\boldsymbol{\sigma}_j^2\|_\infty}{\|\boldsymbol{\sigma}_j^2\|_\infty + n_{j-1}^{\rightarrow}\|\boldsymbol{d}_j^2\|_\infty}$ for $j \geq 2$. For $j = 1$, using the definition of $\boldsymbol{s}_j^{\rightarrow}$ in the

second recursion in (6), we have that for any component $i$

$$\left(s_1^{\to(i)}\right)^{-1} = \left(s_1^{(i)}\right)^{-1} = \frac{n_1}{\sigma_1^{2(i)}} \geq \frac{n_1}{M^2}.$$

Then, vector $s_1^{\to}$ satisfies

$$\|\sqrt{s_1^{\to}}\|_\infty \leq \frac{M}{\sqrt{n_1}} = \frac{M}{\sqrt{n_1^{\to}}}.$$

If $\|\sqrt{s_{j-1}^{\to}}\|_\infty \leq M/\sqrt{n_{j-1}^{\to}}$, then we have that for any component $i$

$$\left(s_j^{\to(i)}\right)^{-1} = \frac{1}{s_j^{(i)}} + \frac{1}{s_{j-1}^{\to(i)} + d_j^{2(i)}} \geq \frac{1}{s_j^{(i)}} + \frac{1}{\frac{M^2}{n_{j-1}^{\to}} + d_j^{2(i)}} \geq \frac{1}{M^2}\left(n_j + \frac{1}{\frac{1}{n_{j-1}^{\to}} + \frac{d_j^{2(i)}}{M^2}}\right)$$

$$\geq \frac{1}{M^2}\left(n_j + \frac{1}{\frac{1}{n_{j-1}^{\to}} + \frac{\|d_j^2\|_\infty}{\|\sigma_j^2\|_\infty}}\right)$$

by using the second recursion in (6) and the induction hypothesis. Then, vector $s_j^{\to}$ satisfies

$$\left\|\sqrt{s_j^{\to}}\right\|_\infty \leq \frac{M}{\sqrt{n_j + n_{j-1}^{\to}\frac{\|\sigma_j^2\|_\infty}{\|\sigma_j^2\|_\infty + n_{j-1}^{\to}\|d_j^2\|_\infty}}}. \tag{17}$$

The inequality in (8) is obtained because the minimax risk is bounded by the smallest minimax risk as shown in [24, 25, 30] so that

$$R(\mathcal{U}_j^{\to}) \leq R_j^\infty + \left(\|\tau_j^\infty - \tau_j^{\to}\|_\infty + \|\lambda_j^{\to}\|_\infty\right)\|\mu_j^\infty\|_1$$

that leads to (8) using (15), (17), and the fact that $1 \leq \sqrt{2\log\left(\frac{2m}{\delta}\right)}$. $\qquad\square$

## D  Proof of Theorem 2

*Proof.* To obtain bound in (9), we proceed by induction. For $j = 1$, using the expression for the ESS in (8), we have that

$$n_1^{\to} = n_1 \geq n.$$

If (9) holds for the $(j-1)$-task, then for the $j$-th task, we have that

$$n_j^{\to} \geq n_j + n_{j-1}^{\to}\frac{\|\sigma_j^2\|_\infty}{\|\sigma_j^2\|_\infty + n_{j-1}^{\to}\|d_j^2\|_\infty} \geq n + n_{j-1}^{\to}\frac{1}{1 + n_{j-1}^{\to}d^2} = n\left(1 + \frac{1}{\frac{n}{n_{j-1}^{\to}} + nd^2}\right)$$

where the second inequality is obtained because $n_j \geq n$, $\|\sigma_j^2\|_\infty \leq 1$, and $\|d_j^2\|_\infty \leq d^2$. Using that $n_{j-1}^{\to} \geq n\left(1 + \frac{(1+\alpha)^{2j-3}-1-\alpha}{\alpha(1+\alpha)^{2j-3}+\alpha}\right)$, the ESS of the $j$-th task satisfies

$$n_j^{\to} \geq n\left(1 + \frac{1}{\frac{n}{n\left(1 + \frac{(1+\alpha)^{2j-3}-1-\alpha}{\alpha(1+\alpha)^{2j-3}+\alpha}\right)} + nd^2}\right) = n\left(1 + \frac{1}{\frac{\alpha(1+\alpha)^{2j-3}+\alpha}{(1+\alpha)^{2j-2}-1} + nd^2}\right)$$

$$= n\left(1 + \frac{1}{\frac{\alpha(1+\alpha)^{2j-3}+\alpha}{(1+\alpha)^{2j-2}-1} + \frac{\alpha^2}{\alpha+1}}\right) \tag{18}$$

$$= n\left(1 + \frac{(1+\alpha)^{2j-1}-1-\alpha}{\alpha(1+\alpha)^{2j-2}+\alpha(\alpha+1)+\alpha^2(1+\alpha)^{2j-2}-\alpha^2}\right)$$

where (18) is obtained because $nd^2 = \frac{\alpha^2}{\alpha+1}$ since $\alpha = \frac{nd^2}{2}\left(\sqrt{1 + \frac{4}{nd^2}} + 1\right)$.

Now, we obtain bounds for the ESS if $nd^2 < \frac{1}{j^2}$. In the following, the constant $\phi$ represents the golden ratio $\phi = 1.618\ldots$.

If $nd^2 < \frac{1}{j^2} \Rightarrow \sqrt{nd^2} \le \alpha \le \sqrt{nd^2}\phi \le \frac{\phi}{j} \le 1$ because $\alpha = \frac{nd^2}{2}\left(\sqrt{1 + \frac{4}{nd^2}} + 1\right) = \sqrt{nd^2}\frac{\sqrt{nd^2+4}+\sqrt{nd^2}}{2}$, then we have that $n_j^{\rightarrow}$ satisfies

$$n_j^{\rightarrow} \ge n\left(1 + \frac{1}{\alpha}\frac{\alpha(2j-2)}{2+\alpha(2j-1)}\right) = n\left(1 + \frac{2j-2}{2+\alpha(2j-1)}\right)$$

where the first inequality follows because $(1+\alpha)^{2j-2} \ge 1 + \alpha(2j-2)$. Using $\alpha \le \frac{\phi}{j}$, we have that

$$n_j^{\rightarrow} \ge n\left(1 + \frac{2j-2}{2 + \frac{\phi}{j}(2j-1)}\right) \ge n\left(1 + \frac{2j-2}{2+2\phi-\frac{\phi}{j}}\right) \ge n\left(1 + \frac{j-1}{1+\phi}\right).$$

$\square$

## E  More efficient recursions for forward and backward learning

The Rauch-Tung-Striebel smoother recursions [34] allow to obtain forward and backward mean and MSE vectors directly from those vectors for the succeeding task. Specifically, for each $j$-th task, the mean vector $\boldsymbol{\tau}_j^{\rightleftharpoons k}$ together with the MSE vector $\boldsymbol{s}_j^{\rightleftharpoons k}$ can be obtained using those vectors for the succeeding task $\boldsymbol{\tau}_{j+1}^{\rightleftharpoons k}, \boldsymbol{s}_{j+1}^{\rightleftharpoons k}$ as

$$\boldsymbol{\tau}_j^{\rightleftharpoons k} = \boldsymbol{\tau}_j^{\rightarrow} + \frac{\boldsymbol{s}_j^{\rightarrow}}{\boldsymbol{s}_j^{\rightarrow} + \boldsymbol{d}_{j+1}^2}\left(\boldsymbol{\tau}_{j+1}^{\rightleftharpoons k} - \boldsymbol{\tau}_j^{\rightarrow}\right)$$

$$\boldsymbol{s}_j^{\rightleftharpoons k} = \left(\frac{1}{\boldsymbol{s}_j^{\rightarrow}} + \left(\boldsymbol{d}_{j+1}^2 + \left(\frac{1}{\boldsymbol{s}_{j+1}^{\rightleftharpoons k}} - \frac{1}{\boldsymbol{s}_j^{\rightarrow} + \boldsymbol{d}_{j+1}^2}\right)^{-1}\right)^{-1}\right)^{-1}.$$

The above recursions provide the same mean vector estimate as the recursions in (10) in the paper since they are obtained using the Rauch-Tung-Striebel smoother recursions instead of fixed-lag smoother recursions [34].

## F  Proof of Theorem 3

*Proof.* To obtain bound in (11) we first prove that the mean vector estimate and the MSE vector given by (10) satisfy

$$\mathbb{P}\left\{|\tau_k^{\infty\,(i)} - \tau_j^{\rightleftharpoons k\,(i)}| \le \kappa\sqrt{2s_j^{\rightleftharpoons k\,(i)}\log\left(\frac{2m}{\delta}\right)}\right\} \ge (1-\delta) \tag{19}$$

for any component $i = 1, 2, \ldots, m$. Then, we prove that $\|\boldsymbol{s}_j^{\rightleftharpoons k}\|_\infty \le M/\sqrt{n_j^{\rightleftharpoons k}}$ for $j \in \{1, 2, \ldots, k\}$, where the ESSs satisfy $n_k^{\rightleftharpoons k} = n_k^{\rightarrow}$ and $n_j^{\rightleftharpoons k} \ge n_j^{\rightarrow} + n_{j+1}^{\leftharpoonup k}\frac{\|\boldsymbol{\sigma}_j^2\|_\infty}{\|\boldsymbol{\sigma}_j^2\|_\infty + n_{j+1}^{\leftharpoonup k}\|\boldsymbol{d}_{j+1}^2\|_\infty}$ for $j \ge 2$.

To obtain inequality (19), we prove that each component $i = 1, 2, \ldots, m$ of the error in the mean vector estimate $z_j^{\rightleftharpoons k\,(i)} = \tau_j^{\infty\,(i)} - \tau_j^{\rightleftharpoons k\,(i)}$ is sub-Gaussian with parameter $\eta_j^{\rightleftharpoons k\,(i)} \le \kappa\sqrt{s_j^{\rightleftharpoons k\,(i)}}$. Analogously to the proof of Theorem 1, it is proven that each component in the error of the backward mean vector $\boldsymbol{\tau}_{j+1}^{\leftharpoonup k}$ is sub-Gaussian with parameters satisfying $\boldsymbol{\eta}_{j+1}^{\leftharpoonup k} \preceq \kappa\sqrt{\boldsymbol{s}_{j+1}^{\leftharpoonup k}}$. The error in the forward and backward mean vector estimate is given by

$$z_j^{\rightleftharpoons k\,(i)} = \tau_j^{\infty\,(i)} - \tau_j^{\rightleftharpoons k\,(i)} = \tau_j^{\infty\,(i)} - \tau_j^{\rightarrow\,(i)} - \frac{s_j^{\rightarrow\,(i)}}{s_j^{\rightarrow\,(i)} + s_{j+1}^{\leftharpoonup k\,(i)} + d_{j+1}^2\,^{(i)}}\left(\tau_{j+1}^{\leftharpoonup k\,(i)} - \tau_j^{\rightarrow\,(i)}\right)$$

where the second equality is obtained using the recursion for $\tau_j^{\rightleftharpoons k\,(i)}$ in (10). Adding and subtracting $\frac{s_j^{\rightarrow(i)}}{s_j^{\rightarrow(i)} + s_{j+1}^{\leftharpoonup k\,(i)} + d_{j+1}^2\,^{(i)}}\tau_{j+1}^{\infty}\,^{(i)}$, we have that

$$z_j^{\rightleftharpoons k\,(i)} = z_j^{\rightarrow(i)} - \frac{s_j^{\rightarrow(i)}}{s_j^{\rightarrow(i)} + s_{j+1}^{\leftharpoonup k\,(i)} + d_{j+1}^2\,^{(i)}}\left(\tau_{j+1}^{\infty}\,^{(i)} - \tau_{j+1}^{\infty}\,^{(i)} + \tau_{j+1}^{\leftharpoonup k\,(i)} - \tau_j^{\rightarrow(i)}\right)$$

$$= z_j^{\rightarrow(i)} - \frac{s_j^{\rightarrow(i)}}{s_j^{\rightarrow(i)} + s_{j+1}^{\leftharpoonup k\,(i)} + d_{j+1}^2\,^{(i)}}\left(\tau_j^{\infty\,(i)} + w_{j+1}^{(i)} - z_{j+1}^{\leftharpoonup k\,(i)} - \tau_j^{\rightarrow(i)}\right)$$

since $\boldsymbol{w}_j = \boldsymbol{\tau}_j^{\infty} - \boldsymbol{\tau}_{j-1}^{\infty}$ and $z_j^{\rightarrow(i)} = \tau_j^{\infty\,(i)} - \tau_j^{\rightarrow(i)}$. Then, we have that

$$z_j^{\rightleftharpoons k\,(i)} = z_j^{\rightarrow(i)} - \frac{s_j^{\rightarrow(i)}}{s_j^{\rightarrow(i)} + s_{j+1}^{\leftharpoonup k\,(i)} + d_{j+1}^2\,^{(i)}}\left(z_j^{\rightarrow(i)} + w_{j+1}^{(i)} - z_{j+1}^{\leftharpoonup k\,(i)}\right) \tag{20}$$

$$= \left(1 - \frac{s_j^{\rightarrow(i)}}{s_j^{\rightarrow(i)} + s_{j+1}^{\leftharpoonup k\,(i)} + d_{j+1}^2\,^{(i)}}\right)z_j^{\rightarrow(i)} - \frac{s_j^{\rightarrow(i)}}{s_j^{\rightarrow(i)} + s_{j+1}^{\leftharpoonup k\,(i)} + d_{j+1}^2\,^{(i)}}\left(w_{j+1}^{(i)} - z_{j+1}^{\leftharpoonup k\,(i)}\right)$$

where $z_j^{\rightarrow(i)}$, $z_{j+1}^{\leftharpoonup k\,(i)}$, and $w_{j+1}^{(i)}$ are sub-Gaussian with parameters $\eta_j^{\rightarrow(i)} \leq \kappa\sqrt{s_j^{\rightarrow(i)}}$, $\eta_{j+1}^{\leftharpoonup k\,(i)} \leq \kappa\sqrt{s_{j+1}^{\leftharpoonup k\,(i)}}$, and $\sigma(w_j^{(i)})$, respectively. Since $\boldsymbol{z}_j^{\rightarrow}$, $\boldsymbol{z}_{j+1}^{\leftharpoonup k}$, and $\boldsymbol{w}_{j+1}$ are independent, we have that $z_j^{\rightleftharpoons k\,(i)}$ given by (20) is sub-Gaussian with parameter that satisfies

$$\left(\eta_j^{\rightleftharpoons k\,(i)}\right)^2 = \left(1 - \frac{s_j^{\rightarrow(i)}}{s_j^{\rightarrow(i)} + s_{j+1}^{\leftharpoonup k\,(i)} + d_{j+1}^2\,^{(i)}}\right)^2\left(\eta_j^{\rightarrow(i)}\right)^2$$

$$+ \left(\frac{s_j^{\rightarrow(i)}}{s_j^{\rightarrow(i)} + s_{j+1}^{\leftharpoonup k\,(i)} + d_{j+1}^2\,^{(i)}}\right)^2\left(\sigma\left(w_j^{(i)}\right)^2 + \left(\eta_{j+1}^{\leftharpoonup k\,(i)}\right)^2\right)$$

$$\leq \left(1 - \frac{s_j^{\rightarrow(i)}}{s_j^{\rightarrow(i)} + s_{j+1}^{\leftharpoonup k\,(i)} + d_{j+1}^2\,^{(i)}}\right)^2\kappa^2 s_j^{\rightarrow(i)}$$

$$+ \left(\frac{s_j^{\rightarrow(i)}}{s_j^{\rightarrow(i)} + s_{j+1}^{\leftharpoonup k\,(i)} + d_{j+1}^2\,^{(i)}}\right)^2\kappa^2\left(d_{j+1}^{(i)} + s_{j+1}^{\leftharpoonup k\,(i)}\right)$$

Using the second recursion in (10) we have that the sub-Gaussian parameter satisfies

$$\left(\eta_j^{\rightleftharpoons k\,(i)}\right)^2 \leq \left(1 - \frac{s_j^{\rightleftharpoons k\,(i)}}{s_{j+1}^{\leftharpoonup k\,(i)} + d_{j+1}^2\,^{(i)}}\right)^2\kappa^2\left(\frac{1}{s_j^{\rightleftharpoons k\,(i)}} - \frac{1}{s_{j+1}^{\leftharpoonup k^2\,(i)} + d_{j+1}^2\,^{(i)}}\right)^{-1}$$

$$+ \frac{\left(s_j^{\rightleftharpoons k\,(i)}\right)^2}{s_{j+1}^{\leftharpoonup k\,(i)} + d_{j+1}^2\,^{(i)}}\kappa^2$$

$$= \left(\frac{s_{j+1}^{\leftharpoonup k\,(i)} + d_{j+1}^2\,^{(i)} - s_j^{\rightleftharpoons k\,(i)}}{s_{j+1}^{\leftharpoonup k\,(i)} + d_{j+1}^2\,^{(i)}}\right)\kappa^2 s_j^{\rightleftharpoons k\,(i)}$$

$$+ \frac{\left(s_j^{\rightleftharpoons k\,(i)}\right)^2}{s_{j+1}^{\leftharpoonup k\,(i)} + d_{j+1}^2\,^{(i)}}\kappa^2 = \kappa^2 s_j^{\rightleftharpoons k\,(i)}.$$

The inequality in (19) is obtained using the union bound together with the Chernoff bound (concentration inequality) [31] for the random variables $z_j^{\rightleftharpoons k\,(i)}$ that are sub-Gaussian with parameter $\eta_j^{\rightleftharpoons k\,(i)}$.

Now, we prove that, for any $j$, $\|\sqrt{s_j^{\rightleftharpoons k}}\| \leq M/\sqrt{n_j^{\rightleftharpoons k}}$ where the ESSs satisfy $n_k^{\rightleftharpoons k} = n_k^{\rightarrow}$ and $n_j^{\rightleftharpoons k} \geq n_j^{\rightarrow} + n_{j+1}^{\leftharpoondown k} \frac{\|\boldsymbol{\sigma}_j^2\|_\infty}{\|\boldsymbol{\sigma}_j^2\|_\infty + n_{j+1}^{\leftharpoondown k}\|\boldsymbol{d}_{j+1}^2\|_\infty}$ for $j \geq 2$. Analogously to the proof of Theorem 1, we prove that the backward MSE vector $s_{j+1}^{\leftharpoondown k}$ satisfies $\|\sqrt{s_{j+1}^{\leftharpoondown k}}\|_\infty \leq M/\sqrt{n_{j+1}^{\leftharpoondown k}}$. Then, using that $\|\sqrt{s_{j+1}^{\leftharpoondown k}}\|_\infty \leq M/\sqrt{n_{j+1}^{\leftharpoondown k}}$, we have that for every component $i$

$$\left(s_j^{\rightleftharpoons k\,(i)}\right)^{-1} = \frac{1}{s_j^{\rightarrow\,(i)}} + \frac{1}{s_{j+1}^{\leftharpoondown k\,(i)} + d_{j+1}^2{}^{(i)}} \geq \frac{n_j^{\rightarrow}}{\sigma_j^{2\,(i)}} + \frac{1}{\frac{M^2}{n_{j+1}^{\leftharpoondown k}} + d_{j+1}^2{}^{(i)}}$$

$$\geq \frac{1}{M^2}\left(n_j^{\rightarrow} + \frac{1}{\frac{1}{n_{j+1}^{\leftharpoondown k}} + \frac{d_{j+1}^2}{M^2}}\right) \geq \frac{1}{M^2}\left(n_j^{\rightarrow} + \frac{1}{\frac{1}{n_{j+1}^{\leftharpoondown k}} + \frac{\|d_{j+1}^2\|_\infty}{\|\boldsymbol{\sigma}_j^2\|_\infty}}\right).$$

Then, we obtain

$$\|\sqrt{s_j^{\rightleftharpoons k}}\|_\infty \leq \frac{M}{\sqrt{n_j^{\rightarrow} + \frac{1}{\frac{1}{n_{j+1}^{\leftharpoondown k}} + \frac{\|d_{j+1}^2\|_\infty}{\|\boldsymbol{\sigma}_j^2\|_\infty}}}}. \tag{21}$$

The inequality in (11) is obtained because the minimax risk is bounded by the smallest minimax risk as shown in [24, 25, 30] so that

$$R(\mathcal{U}_j^{\rightleftharpoons k}) \leq R_j^\infty + \left(\|\boldsymbol{\tau}_j^\infty - \boldsymbol{\tau}_j^{\rightleftharpoons k}\|_\infty + \|\boldsymbol{\lambda}_j^{\rightleftharpoons k}\|_\infty\right)\|\boldsymbol{\mu}_j^\infty\|_1$$

that leads to (11) using (19), (21), and the fact that $1 \leq \sqrt{2\log\left(\frac{2m}{\delta}\right)}$. $\qquad\square$

## G  Proof of Theorem 4

*Proof.* To obtain bound in (12), we use the ESS obtained with forward learning in Theorem 2 and obtained with backward learning. Analogously to the proof of Theorem 2, we prove that the ESS obtained at backward learning satisfies

$$n_{j+1}^{\leftharpoondown k} \geq n_{j+1} + n_{j+2}^{\leftharpoondown k}\frac{\|\boldsymbol{\sigma}_{j+1}^2\|_\infty}{\|\boldsymbol{\sigma}_{j+1}^2\|_\infty + n_{j+2}^{\leftharpoondown k}\|\boldsymbol{d}_{j+2}^2\|_\infty} \geq n\left(1 + \frac{(1+\alpha)^{2(k-j)-1} - 1 - \alpha}{\alpha(1+\alpha)^{2(k-j)-1} + \alpha}\right).$$

Therefore, the ESS obtained with forward an backward learning satisfies

$$n_j^{\rightleftharpoons k} \geq n_j^{\rightarrow} + n\left(1 + \frac{(1+\alpha)^{2(k-j)-1} - 1 - \alpha}{\alpha(1+\alpha)^{2(k-j)-1} + \alpha}\right)\left(1 + \frac{n\left(1 + \frac{(1+\alpha)^{2(k-j)-1} - 1 - \alpha}{\alpha(1+\alpha)^{2(k-j)-1} + \alpha}\right)}{nd^2}\right)^{-1}$$

$$= n_j^{\rightarrow} + n\frac{(1+\alpha)^{2(k-j)} - 1}{\alpha(1+\alpha)^{2(k-j)-1} + \alpha}\left(1 + \frac{\alpha^2}{\alpha+1}\left(1 + \frac{(1+\alpha)^{2(k-j)-1} - 1 - \alpha}{\alpha(1+\alpha)^{2(k-j)-1} + \alpha}\right)\right)^{-1}$$

where the second equality follows because $nd^2 = \frac{\alpha^2}{\alpha+1}$ since $\alpha = \frac{nd^2}{2}\left(\sqrt{1 + \frac{4}{nd^2}} + 1\right)$. Then, we have that

$$n_j^{\rightleftharpoons k} \geq n_j^{\rightarrow} + n\frac{(1+\alpha)^{2(k-j)} - 1}{\alpha(1+\alpha)^{2(k-j)-1} + \alpha}$$

$$\cdot\left(\frac{((1+\alpha)^{2(k-j)-1} + 1)(\alpha + 1 + \alpha^2) + \alpha((1+\alpha)^{2(k-j)-1} - 1 - \alpha)}{(\alpha+1)((1+\alpha)^{2(k-j)-1} + 1)}\right)^{-1}$$

$$\geq n_j^{\rightarrow} + n\frac{(1+\alpha)^{2(k-j)} - 1}{\alpha(1+\alpha)^{2(k-j)-1} + \alpha}\frac{(\alpha+1)((1+\alpha)^{2(k-j)-1} + 1)}{(1+\alpha)^{2(k-j)+1} + 1}.$$

Now, we obtain bounds for the ESS depending on the value value of $nd^2$. Such bounds are obtained similarly as in Theorem 2 and we also denote by $\phi$ the golden ratio $\phi = 1.618\ldots$.

1. If $nd^2 < \frac{1}{j^2} \Rightarrow \sqrt{nd^2} \leq \alpha \leq \sqrt{nd^2}\phi \leq \frac{\phi}{j} \leq 1$ because

$$\alpha = nd^2\frac{\sqrt{1 + \frac{4}{nd^2}} + 1}{2} = \sqrt{nd^2}\frac{\sqrt{nd^2 + 4} + \sqrt{nd^2}}{2}$$

then we have that $n_j^{\rightleftharpoons k}$ satisfies

$$n_j^{\rightleftharpoons k} \geq n_j^{\rightarrow} + n\frac{1}{\alpha}\frac{\alpha(2(k-j))}{2 + \alpha 2(k-j)} = n_j^{\rightarrow} + n\frac{k-j}{1 + \alpha(k-j)} \geq n_j^{\rightarrow} + n\frac{k-j}{1 + \frac{\phi}{j}(k-j)}$$

where the first inequality follows because $(1+\alpha)^{2(k-j)-1} \geq 1 + \alpha(2(k-j)-1)$ and the second inequality is obtained using $\alpha \leq \frac{\phi}{j}$.

2. If $\frac{1}{j^2} \leq nd^2 < 1 \Rightarrow \frac{1}{j} \leq \sqrt{nd^2} \leq \alpha \leq \sqrt{nd^2}\phi$ because

$$\alpha = nd^2\frac{\sqrt{1 + \frac{4}{nd^2}} + 1}{2} = \sqrt{nd^2}\frac{\sqrt{nd^2 + 4} + \sqrt{nd^2}}{2}$$

then we have that $n_j^{\rightleftharpoons k}$ satisfies

$$n_j^{\rightleftharpoons k} \geq n_j^{\rightarrow}\frac{n}{\alpha}\frac{(1+\alpha)^{2(k-j)} - 1}{(1+\alpha)^{2(k-j)} + 1} \geq n_j^{\rightarrow}\frac{n}{\alpha}\frac{(1+\sqrt{nd^2})^{2(k-j)} - 1}{(1+\sqrt{nd^2})^{2(k-j)} + 1}$$

where the second inequality follows because the ESS is monotonically increasing for $\alpha$ and $\alpha \geq nd^2$. Since $(1 + \sqrt{nd^2})^{2(k-j)} \geq 1 + 2\sqrt{nd^2}(k-j)$ and $k - j \geq 1$, we have that

$$n_j^{\rightleftharpoons k} \geq n_j^{\rightarrow} + \frac{n}{\alpha}\frac{\sqrt{nd^2}}{1 + \sqrt{nd^2}} \geq n_j^{\rightarrow} + n\frac{1}{\phi}\frac{1}{1 + \sqrt{nd^2}}$$

because $\alpha \leq \sqrt{nd^2}\phi$.

3. If $nd^2 \geq 1 \Rightarrow 1 \leq nd^2 \leq \alpha \leq nd^2\phi$ because $\alpha = nd^2\frac{\sqrt{1 + \frac{4}{nd^2}} + 1}{2}$, then we have that $n_j^{\rightleftharpoons k}$ satisfies

$$n_j^{\rightleftharpoons k} \geq n_j^{\rightarrow} + n\frac{1}{\alpha}\frac{2^{2(k-j)} - 1}{2^{2(k-j)} + 1} \geq n_j^{\rightarrow} + n\frac{1}{nd^2}\frac{1}{\phi}\frac{3}{5}$$

where the first inequality follows because the ESS is monotonically increasing for $\alpha$ and $\alpha \geq 1$ and the second inequality is obtained using $k - j \geq 1$ and $\alpha \leq nd^2\phi$.

$\square$

## H  Additional numerical results and implementation details

In this section we describe the datasets used for the numerical results in Section 5, we provide further details for the numerical experimentations carried out, and include several additional results. Specifically, in the first set of additional results, we evaluate the classification performance of the proposed method in comparison with state-of-the-art techniques for different sample sizes; in the second set of additional results, we further show the performance improvement leveraging information from all the tasks in the sequence with additional datasets; in the third set of additional results, we show the classification error and the running time of IMRCs for different hyper-parameter values; and in the fourth set of additional results, we evaluate the assumption of change between tasks being independent and zero-mean. In addition, in the folder Implementation_IMRC in the supplementary materials we provide the code of the proposed IMRCs with the setting used in the numerical results.

In Section 5, we use 12 publicly available datasets [1, 37, 36, 39, 10, 40, 41, 2], and http://yann.lecun.com/exdb/mnist/. The summary of these datasets is provided in Table 3 that shows the number of classes, the number of samples, and the number of tasks. In the following, we further describe the tasks and the time-dependency of each dataset used.

- The "Yearbook" dataset contains portraits' photographs over time and the goal is to predict males and females. Each task corresponds to portraits from one year from 1905 to 2013.
- The "ImageNet noise" dataset contains images with increasing noise over tasks and the goal is to predict if an image is a bird or a snake. The sequence of tasks corresponds to the noise factors $[0.0, 0.4, 0.8, 1.2, 1.6, 2.0, 2.4, 2.8, 3.2, 3.6]$ [35].
- The "DomainNet" dataset contains six different domains with decreasing realism and the goal is to predict if an image is an airplane, bus, ambulance, or police car. The sequence of tasks corresponds to the six domains: real, painting, infograph, clipart, sketch, and quickdraw.
- The "UTKFaces" dataset contains face images in the wild with increasing age and the goal is to predict males and females. The sequence of tasks corresponds to face images with different ages from 0 to 116 years.
- The "Rotated MNIST" dataset contains rotated images with increasing angles over tasks and the goal is to predict if the number in an image is greater than $5$ or not. Each $j$-th task corresponds to a rotation angle randomly selected from $\left[ \frac{180(j-1)}{k}, \frac{180j}{k} \right]$ degrees where $j \in \{1, 2, \ldots, k\}$ and $k$ is the number of tasks.
- The "CLEAR" dataset contains images with a natural temporal evolution of visual concepts and the goal is to predict if an image is soccer, hockey, or racing. Each task corresponds to one year from 2004 to 2014.
- The "Power Supply" dataset contains three year power supply records from 1995 to 1998 and the goal is to predict which hour the current power supply belongs to. We relabel into binary classification according to pm. or am. as in [10].
- The "Usenet" dataset is splitted into Usenet1 and Usenet2 which both contains a stream of messages from different 20 newsgroups that are sequentially presented to a user and the goal is to predict the personal interests.
- The "German" dataset contains information about people who take a credit by a bank and the goal is to classify each person as good or bad credit risks.
- The "Spam" dataset contains emails and the task is to predict if an email is malicious spam email or legitimate email.
- The "Covertype" dataset contains cartographic variables of a forest area obtained from US Forest Service (USFS) and the goal is to predict the forest cover type.

The samples in each task are randomly splitted in 100 samples for test and the rest of the samples for training. The samples used for training in the numerical results are randomly sampled from each group of training samples in each repetition.

In Section 5, we compare the results of IMRC methods with 7 state-of-the-art-techniques [5, 17, 4, 6, 9, 10, 16]. In the following, we briefly describe each method used.

- GEM method [5] is a technique developed for continual learning. The method provided by Lopez-Paz & Ranzato learns each new task using a stochastic gradient descent with inequality constraints given by the losses of preceding tasks. Such constraints avoid the increase of the loss of each preceding tasks.
- MER method [17] is a technique developed for continual learning. The method provided by Riemer et al. learns each new task using sample sample sets that include random samples of preceding tasks. Such samples of preceding tasks are stored in a memory buffer.
- ELLA method [4] is a techniques developed for continual learning. The method provided by Ruvolo & Eaton learns each new task transferring knowledge from a shared basis of task models.
- EWC method [6] is a technique developed for continual learning. The method provided by Kirkpatrick et al. learns each new task regularizing the loss with regularization parameters given by the Fisher information.
- Condor method [9] is a technique developed for concept drift adaptation. The method provided by Zhao et al. is an ensemble method that adapts to evolving tasks by learning weighting the models in the ensemble at each time step.

Table 3: Datasets characteristics.

| Dataset | Classes | Samples | Tasks |
|---|---|---|---|
| Yearbook [1] | 2 | 37,921 | 126 |
| ImageNet Noise [35] | 2 | 12,000 | 10 |
| DomainNet [36] | 4 | 6,256 | 6 |
| UTKFace [37] | 2 | 23,500 | 94 |
| Rotated MNIST [38] | 2 | 70,000 | 60 |
| CLEAR [39] | 3 | 10,490 | 10 |
| Power Supply [10] | 2 | 29,928 | 99 |
| Usenet1 [40] | 2 | 1,500 | 5 |
| Usenet2 [40] | 2 | 1,500 | 5 |
| German [41] | 2 | 1,000 | 3 |
| Spam [2] | 2 | 6,213 | 20 |
| Covertype [10] | 2 | 581,012 | 1,936 |

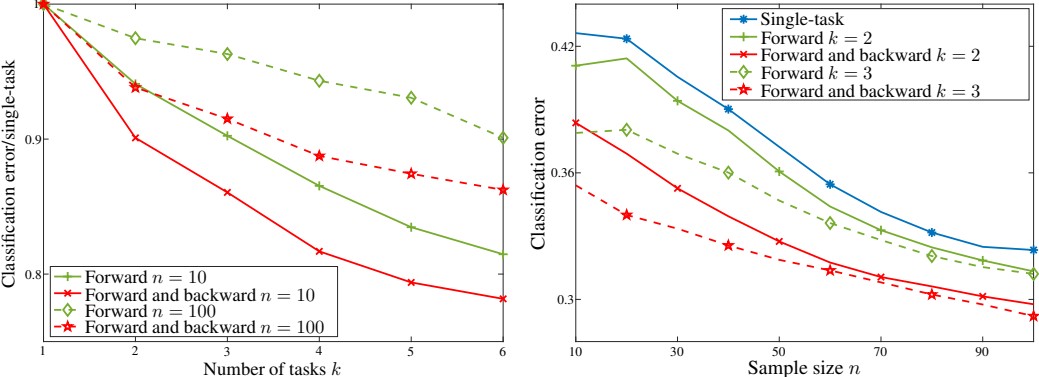

(a) Classification error per number of tasks using "DomainNet" dataset.

(b) Classification error per sample size using "DomainNet" dataset.

Figure 5: Forward and backward learning can sharply boost performance and ESS as tasks arrive.

- DriftSurf method [10] is a technique developed for concept drift adaptation. The method provided by Tahmasbi et al. adapts to evolving tasks by using a drift detection method. Such method allows to restart a new model when a change in the distribution is detected.
- AUE method [16] is a technique developed for concept drift adaptation. The method provided by Brzezinski & Stefanowski is an ensemble method that adapts to evolving tasks by incrementally updating all classifiers in the ensemble and weighting them with non-linear error functions.

The classifier parameters in the numerical results are obtained using an accelerated subgradient method based on Nesterov approach [32, 33]. Such subgradient method applied to optimization (3) obtains at each step classifier parameters $\boldsymbol{\mu}$ from the mean and confidence vectors $\boldsymbol{\tau}, \boldsymbol{\lambda}$ using the iterations for $l = 1, 2, \ldots, K$

$$\bar{\boldsymbol{\mu}}(l+1) = \boldsymbol{\mu}(l) + a_l\Big(\boldsymbol{\tau} - \partial\varphi(\boldsymbol{\mu}(l)) - \boldsymbol{\lambda}\text{sign}(\boldsymbol{\mu}(l))\Big) \quad (22)$$

$$\boldsymbol{\mu}(l+1) = \bar{\boldsymbol{\mu}}(l+1) + \theta_{l+1}(\theta_l^{-1} - 1)\left(\boldsymbol{\mu}(l) - \bar{\boldsymbol{\mu}}(l)\right)$$

where $\text{sign}(\cdot)$ denotes the sign function, $\boldsymbol{\mu}(l)$ is the $l$-th iterate for $\boldsymbol{\mu}$, $\theta_l = 2/(l+1)$ and $a_l = 1/(l+1)^{3/2}$ are the step sizes and $\partial\varphi(\boldsymbol{\mu}(l))$ denotes a subgradient of $\varphi(\cdot)$ at $\boldsymbol{\mu}(l)$ with

$$\varphi(\boldsymbol{\mu}) = \max_{x \in \mathcal{X}, \mathcal{C} \subseteq \mathcal{Y}} \frac{\sum_{y \in \mathcal{C}} \Phi(x,y)^{\mathrm{T}}\boldsymbol{\mu} - 1}{|\mathcal{C}|}.$$

In addition, the above subgradient method is implemented using $K = 2000$ iterations and a warm-start that initializes the classifier parameters in (22) with the solution obtained for the closest task.

Table 4: Classification error and standard deviation of the proposed IMRC method in comparison with the existing techniques for $n = 50$ samples per task.

| Method | GEM | MER | ELLA | EWC | Condor | DriftSurf | AUE | IMRC |
|---|---|---|---|---|---|---|---|---|
| Yearbook | .16 ± .02 | .11 ± .01 | .43 ± .10 | .38 ± .02 | .11 ± .02 | .29 ± .02 | .29 ± .02 | **.10 ± .02** |
| I. noise | .16 ± .06 | **.10 ± .01** | .47 ± .05 | .47 ± .05 | **.10 ± .02** | .48 ± .02 | .48 ± .02 | **.10 ± .02** |
| DomainNet | .65 ± .05 | .29 ± .02 | .67 ± .05 | .75 ± .05 | .45 ± .01 | .32 ± .00 | .32 ± .00 | **.28 ± .03** |
| UTKFaces | .12 ± .00 | .11 ± .09 | .18 ± .11 | .12 ± .00 | .14 ± .02 | .12 ± .00 | .12 ± .00 | **.10 ± .00** |
| R. MNIST | .29 ± .04 | .41 ± .13 | .48 ± .05 | .44 ± .01 | .43 ± .01 | .48 ± .02 | .48 ± .02 | **.25 ± .02** |
| CLEAR | .08 ± .01 | .09 ± .04 | .60 ± .05 | .64 ± .03 | .35 ± .01 | .33 ± .00 | .33 ± .00 | **.05 ± .02** |
| P. Supply | .47 ± .03 | .47 ± .02 | .35 ± .01 | .47 ± .02 | .36 ± .01 | .40 ± .01 | .40 ± .01 | **.28 ± .01** |
| Usenet1 | .47 ± .02 | .46 ± .02 | .32 ± .01 | .48 ± .02 | .45 ± .02 | .39 ± .01 | .39 ± .01 | **.30 ± .01** |
| Usenet2 | .35 ± .01 | .32 ± .02 | .32 ± .01 | .42 ± .05 | .35 ± .02 | **.27 ± .01** | .30 ± .01 | .32 ± .01 |
| German | .34 ± .01 | .33 ± .01 | .31 ± .01 | .30 ± .01 | .34 ± .01 | .30 ± .01 | .30 ± .01 | **.28 ± .01** |
| Spam | **.07 ± .01** | .09 ± .02 | .23 ± .03 | .23 ± .03 | .19 ± .01 | .32 ± .01 | .33 ± .01 | .08 ± .00 |
| Covert. | .09 ± .00 | .08 ± .00 | .11 ± .01 | .08 ± .00 | **.07 ± .02** | .09 ± .00 | .10 ± .00 | .08 ± .00 |

Table 5: Classification error and standard deviation of the proposed IMRC method in comparison with the existing techniques for $n = 100$ samples per task.

| Method | GEM | MER | ELLA | EWC | Condor | DriftSurf | AUE | IMRC |
|---|---|---|---|---|---|---|---|---|
| Yearbook | .17 ± .03 | .10 ± .01 | .43 ± .02 | .27 ± .06 | .09 ± .01 | .27 ± .02 | .27 ± .01 | **.08 ± .02** |
| I. noise | .13 ± .07 | .10 ± .01 | .47 ± .04 | .46 ± .06 | .09 ± .01 | .48 ± .02 | .48 ± .02 | **.09 ± .01** |
| DomainNet | .53 ± .10 | **.26 ± .04** | .67 ± .05 | .74 ± .05 | .44 ± .00 | .32 ± .00 | .32 ± .00 | .28 ± .01 |
| UTKFaces | .12 ± .00 | .11 ± .01 | .17 ± .11 | .12 ± .00 | .11 ± .01 | .12 ± .00 | .12 ± .00 | **.10 ± .00** |
| R. MNIST | .28 ± .02 | .45 ± .10 | .47 ± .05 | .40 ± .01 | .41 ± .01 | .48 ± .02 | .48 ± .02 | **.21 ± .00** |
| CLEAR | .09 ± .02 | **.05 ± .02** | .60 ± .05 | .62 ± .04 | .35 ± .01 | .33 ± .00 | .34 ± .00 | **.05 ± .02** |
| P. Supply | .47 ± .02 | .48 ± .03 | .37 ± .04 | .46 ± .00 | .36 ± .00 | .37 ± .02 | .38 ± .02 | .25 ± .00 |
| Usenet1 | .46 ± .02 | .46 ± .02 | .35 ± .05 | .47 ± .02 | .40 ± .01 | .39 ± .01 | .39 ± .01 | .30 ± .01 |
| Usenet2 | .34 ± .00 | .29 ± .01 | .35 ± .02 | .35 ± .02 | .32 ± .02 | **.26 ± .02** | .27 ± .02 | .30 ± .00 |
| German | .34 ± .01 | **.29 ± .01** | **.29 ± .01** | .30 ± .01 | .30 ± .01 | **.29 ± .00** | .32 ± .01 | **.29 ± .00** |
| Spam | **.07 ± .01** | .08 ± .01 | .26 ± .05 | .17 ± .02 | .10 ± .02 | .32 ± .01 | .33 ± .01 | .08 ± .00 |
| Covert. | .08 ± .00 | .08 ± .00 | .11 ± .00 | .08 ± .00 | **.07 ± .02** | .09 ± .00 | .09 ± .00 | .08 ± .00 |

Table 6: Classification error and standard deviation of the proposed IMRC method in comparison with the existing techniques for $n = 150$ samples per task.

| Method | GEM | MER | ELLA | EWC | Condor | DriftSurf | AUE | IMRC |
|---|---|---|---|---|---|---|---|---|
| Yearbook | .16 ± .03 | .10 ± .01 | .43 ± .08 | .22 ± .02 | .09 ± .01 | .23 ± .01 | .23 ± .01 | **.08 ± .01** |
| I. noise | .12 ± .03 | **.07 ± .01** | .47 ± .04 | .45 ± .07 | .09 ± .01 | .48 ± .02 | .48 ± .02 | .08 ± .01 |
| DomainNet | .49 ± .10 | .28 ± .02 | .67 ± .05 | .74 ± .05 | .43 ± .01 | .32 ± .00 | .32 ± .00 | **.27 ± .02** |
| UTKFaces | .12 ± .00 | .11 ± .01 | .17 ± .11 | .12 ± .00 | **.10 ± .01** | .12 ± .00 | .12 ± .00 | **.10 ± .00** |
| R. MNIST | .27 ± .01 | .47 ± .05 | .47 ± .05 | .38 ± .01 | .41 ± .02 | .48 ± .02 | .48 ± .02 | **.20 ± .01** |
| CLEAR | .08 ± .01 | .05 ± .02 | .60 ± .04 | .60 ± .04 | .36 ± .01 | .33 ± .00 | .33 ± .00 | **.04 ± .01** |
| P. Supply | .47 ± .01 | .48 ± .02 | .30 ± .01 | .47 ± .01 | .36 ± .01 | .36 ± .00 | .36 ± .01 | **.24 ± .00** |
| Usenet1 | .47 ± .02 | .41 ± .03 | **.29 ± .01** | .48 ± .01 | .43 ± .02 | .39 ± .00 | .39 ± .01 | **.29 ± .01** |
| Usenet2 | .34 ± .00 | .29 ± .00 | **.26 ± .01** | .32 ± .01 | .32 ± .02 | **.26 ± .00** | .28 ± .00 | .31 ± .00 |
| German | .34 ± .01 | .30 ± .00 | **.25 ± .01** | .30 ± .02 | .31 ± .01 | .29 ± .01 | .29 ± .01 | .28 ± .01 |
| Spam | **.07 ± .01** | **.07 ± .01** | .23 ± .03 | .13 ± .02 | .11 ± .02 | .32 ± .01 | .33 ± .01 | **.07 ± .00** |
| Covert. | .08 ± .00 | .08 ± .00 | .10 ± .01 | .08 ± .00 | **.07 ± .02** | .09 ± .00 | .09 ± .00 | .08 ± .00 |

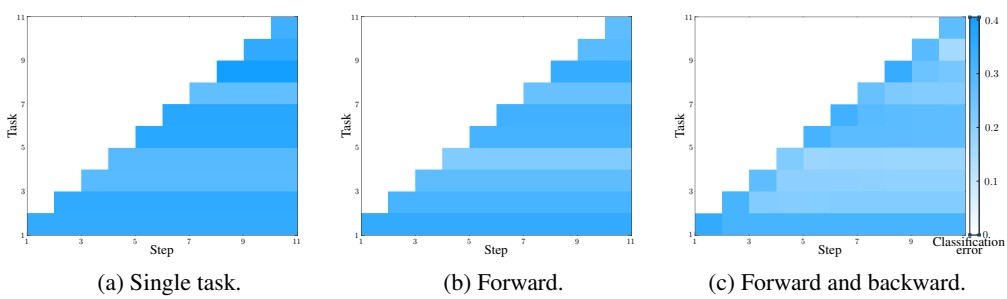

|  | (a) Single task. | (b) Forward. | (c) Forward and backward. |

Figure 6: Forward and backward learning can improve performance of preceding tasks.

Table 7: Classification error of the proposed IMRC method varying $W$ and $b$.

| Hyper-parameter | $W = 2$ | | $W = 4$ | | $W = 6$ | | $b = 1$ | | $b = 2$ | | $b = 3$ | | $b = 4$ | | $b = 5$ | |
|---|---|---|---|---|---|---|---|---|---|---|---|---|---|---|---|---|
| Sample size $n$ | 10 | 100 | 10 | 100 | 10 | 100 | 10 | 100 | 10 | 100 | 10 | 100 | 10 | 100 | 10 | 100 |
| Yearbook | .13 | .08 | .13 | .09 | .13 | .09 | .14 | .10 | .09 | .14 | .13 | .08 | .13 | .08 | .13 | .08 |
| ImageNet noise | .15 | .09 | .15 | .09 | .15 | .09 | .15 | .09 | .15 | .09 | .15 | .09 | .15 | .08 | .15 | .08 |
| DomainNet | .34 | .28 | .32 | .27 | .33 | .28 | .36 | .29 | .35 | .28 | .34 | .28 | .34 | .28 | .34 | .28 |
| UTKFaces | .10 | .10 | .10 | .10 | .10 | .10 | .10 | .10 | .10 | .10 | .10 | .10 | .10 | .10 | .10 | .10 |
| Rotated MNIST | .36 | .21 | .35 | .21 | .36 | .21 | .36 | .22 | .36 | .22 | .36 | .21 | .36 | .21 | .36 | .21 |
| CLEAR | .09 | .05 | .09 | .05 | .09 | .06 | .10 | .05 | .09 | .05 | .09 | .05 | .09 | .05 | .08 | .05 |
| Weather | .31 | .31 | .31 | .31 | .31 | .31 | .31 | .31 | .31 | .31 | .31 | .31 | .31 | .31 | .31 | .31 |
| Power Supply | .30 | .25 | .30 | .25 | .30 | .25 | .32 | .27 | .31 | .26 | .30 | .25 | .30 | .25 | .30 | .25 |
| Usenet1 | .32 | .30 | .33 | .31 | .33 | .31 | .32 | .30 | .32 | .30 | .32 | .30 | .32 | .30 | .32 | .30 |
| Usenet2 | .33 | .30 | .34 | .31 | .34 | .31 | .34 | .30 | .33 | .30 | .33 | .30 | .33 | .30 | .32 | .30 |
| German | .34 | .29 | .34 | .29 | .34 | .29 | .34 | .29 | .34 | .29 | .34 | .29 | .34 | .29 | .34 | .29 |
| Spam | .13 | .08 | .14 | .08 | .13 | .08 | .14 | .09 | .13 | .08 | .13 | .08 | .12 | .07 | .12 | .07 |
| Covertype | .08 | .08 | .08 | .08 | .08 | .08 | .08 | .08 | .08 | .08 | .08 | .08 | .08 | .08 | .08 | .08 |

In the **first set of additional results**, we further compare the classification error of IMRCs with the state-of-the-art techniques. The results in Table 1 in the paper as well as Tables 4, 5, and 6 are obtained computing the classification error 50 times for each sample size. Table 1 in the paper shows classification errors for $n = 10$ samples, while Tables 4, 5, and 6 show the classification error for $n = 50$, $n = 100$, and $n = 150$ samples, respectively. As can be observed in Table 4, the performance improvement of IMRCs in comparison with the state-of-the-art techniques for $n = 50, n = 100$, and $n = 150$ is similar to that shown in the paper for $n = 10$.

In the **second set of additional results**, we further illustrate the relationship among classification error, number of tasks, and sample size. Figure 4 in the paper as well as Figure 5 are obtained computing the classification error over all the sequences of consecutive tasks of length $k$ in the dataset. Then, we repeat such experiment 10 times with randomly chosen training sets of size $n$. Figure 5 extends the results for IMRCs using "DomainNet" dataset completing those in the main paper that show the results using "Yearbook" dataset. Figure 5a shows the classification error of IMRC method divided by the classification error of single-task learning for different number of tasks with $n = 10$ and $n = 100$ sample sizes. In addition, Figure 5b shows the classification error of IMRC method for different sample sizes with $k = 10$ tasks. Figures 5a and 5b show similar behavior to Figures 4a and 4b in the paper, respectively. In addition, Figure 6 shows the classification error of IMRCs per step and task with single-task learning, forward learning, and forward and backward learning using the "Yearbook" dataset. Such figure shows that forward and backward learning can improve performance of preceding tasks, while forward learning and single task learning maintain the same performance over time.

In the **third set of additional results**, we further assess the change in classification error and the running time of IMRCs varying the hyper-parameters. Table 7 shows the classification error of IMRCs varying the values of hyper-parameter for the window size $W$ and the number of backward steps $b$, completing those in the paper that show the results for $W = 2$ and $b = 3$. As shown in the table, the proposed IMRCs do not require a careful fine-tuning of hyper-parameters and similar performances are obtained by using different values. In addition, Table 8 shows the mean running

Table 8: Running time of IMRC method in comparison with the state-of-the-art-techniques.

| Dataset | GEM | | MER | | ELLA | | EWC | | $b=1$ | | $b=2$ | | $b=3$ | | $b=4$ | | $b=5$ | |
|---|---|---|---|---|---|---|---|---|---|---|---|---|---|---|---|---|---|---|
| Sample size $n$ | 10 | 100 | 10 | 100 | 10 | 100 | 10 | 100 | 10 | 100 | 10 | 100 | 10 | 100 | 10 | 100 | 10 | 100 |
| Yearbook | 0.098 | 0.476 | 0.166 | 3.726 | 0.066 | 0.070 | 0.358 | 3.031 | 0.094 | 0.324 | 0.105 | 0.397 | 0.133 | 0.487 | 0.167 | 0.582 | 0.184 | 0.664 |
| ImageNet noise | 0.010 | 0.037 | 0.079 | 1.052 | 0.073 | 0.073 | 0.032 | 0.252 | 0.259 | 0.490 | 0.261 | 0.531 | 0.284 | 0.561 | 0.304 | 0.585 | 0.438 | 0.601 |
| DomainNet | 0.005 | 0.020 | 0.066 | 0.900 | 0.054 | 0.065 | 0.018 | 0.155 | 0.518 | 8.463 | 0.543 | 8.983 | 0.542 | 9.514 | 0.559 | 9.699 | 0.571 | 9.921 |
| UTKFaces | 0.310 | 0.180 | 0.127 | 3.458 | 0.059 | 0.062 | 0.245 | 2.246 | 0.108 | 0.348 | 0.115 | 0.401 | 0.133 | 0.488 | 0.156 | 0.573 | 0.190 | 0.664 |
| Rotated MNIST | 0.180 | 1.094 | 0.211 | 4.092 | 0.176 | 0.198 | 0.587 | 5.296 | 0.135 | 0.471 | 0.165 | 0.599 | 0.209 | 0.737 | 0.249 | 0.877 | 0.288 | 1.010 |
| CLEAR | 0.009 | 0.034 | 0.074 | 1.063 | 0.077 | 0.073 | 0.031 | 0.248 | 0.235 | 1.310 | 0.252 | 1.406 | 0.255 | 1.469 | 0.271 | 1.595 | 0.360 | 1.693 |
| Power Supply | 0.031 | 0.169 | 0.025 | 0.249 | 0.007 | 0.006 | 0.190 | 1.937 | 0.241 | 0.768 | 0.257 | 0.840 | 0.291 | 1.025 | 0.351 | 1.209 | 0.425 | 1.404 |
| Usenet1 | 0.003 | 0.014 | 0.012 | 0.115 | 0.008 | 0.009 | 0.012 | 0.117 | 0.320 | 0.930 | 0.324 | 0.934 | 0.366 | 0.946 | 0.368 | 0.980 | 0.400 | 1.472 |
| Usenet2 | 0.003 | 0.013 | 0.012 | 0.113 | 0.008 | 0.009 | 0.012 | 0.117 | 0.272 | 1.164 | 0.288 | 1.194 | 0.302 | 1.224 | 0.306 | 1.218 | 0.322 | 1.772 |
| German | 0.002 | 0.009 | 0.011 | 0.114 | 0.010 | 0.011 | 0.010 | 0.074 | 0.260 | 0.770 | 0.266 | 0.773 | 0.303 | 0.773 | 0.466 | 1.110 | 0.606 | 1.796 |
| Spam | 0.014 | 0.061 | 0.020 | 0.194 | 0.060 | 0.062 | 0.055 | 0.552 | 0.235 | 1.061 | 0.265 | 1.090 | 0.267 | 1.242 | 0.295 | 1.278 | 0.321 | 2.020 |
| Covertype | 1.094 | 3.124 | 2.194 | 6.431 | 0.057 | 0.060 | 2.201 | 6.438 | 0.139 | 0.586 | 0.166 | 0.697 | 0.211 | 0.850 | 0.244 | 1.003 | 0.278 | 1.15 |

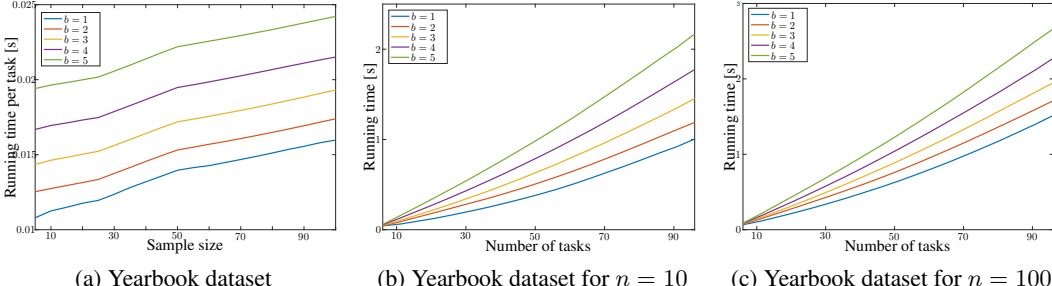

(a) Yearbook dataset   (b) Yearbook dataset for $n = 10$   (c) Yearbook dataset for $n = 100$

Figure 7: Running time per task varying the sample size, the number of tasks, and the number of backward steps.

time per task in seconds of IMRCs for $b = 1, 2, \ldots, 5$ backward steps in comparison with the state-of-the-art techniques that learn a sequence of tasks. Such table shows that the methods proposed for backward learning do not require a significant increase in complexity, and the running time of the proposed method is similar to that of other state-of-the-art methods. Figure 7 further assesses the running time of IMRCs for $b = 1, 2, \ldots, 5$ backward steps varying the sample size and the number of tasks. Such figure shows that the running time of IMRCs increases moderately and linearly with the number of backward steps and the number of tasks.

In the **fourth set of additional results**, we evaluate the assumption of change between tasks being independent and zero-mean by assessing the partial autocorrelation of mean vectors. In particular, the partial autocorrelation at any lag would be zero if tasks are i.i.d.; while the partial autocorrelation at lag 1 is larger than zero if tasks satisfy the assumption of Section 2. Figure 8 shows the averaged partial autocorrelation of the mean vectors components +/- their standard deviations for different lags using "Yearbook" and "UTKFaces" datasets. Such figure shows a partial autocorrelation clearly non-zero at lag 1 that reflects dependence between consecutive mean vectors, as described by the assumption of Section 2.

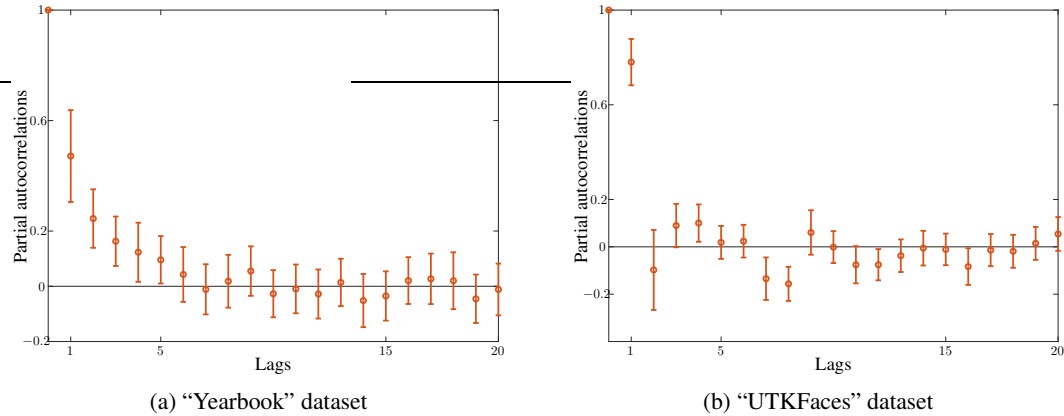

(a) "Yearbook" dataset   (b) "UTKFaces" dataset

Figure 8: Averaged partial autocorrelation of mean vectors components +/- their standard deviations.

