# OpenReview forum: "Minimax Forward and Backward Learning of Evolving Tasks with Performance Guarantees"
_NeurIPS.cc/2023/Conference — NeurIPS 2023 poster_

### Official Review · Reviewer_C5UC · 2023-07-06

**Soundness:** 2 fair
**Presentation:** 3 good
**Contribution:** 3 good
**Rating:** 5
**Confidence:** 5

**Summary:**

This paper presents an incremental minimax risk classifier (imrc) to effectively utilize forward and reverse learning and explain evolutionary tasks. Furthermore, the performance improvements provided by forward and backward learning can also be described analytically based on the expected quadratic changes of the task and the number of tasks.

**Strengths:**

In the article, the effects and limitations of IMRCs are verified by rich experiments, and using IMRCs can better use forward and reverse learning to explain the evolutionary tasks and improve the effective sample size.

**Weaknesses:**

However, there are some deficiencies in the ablation aspect, and the experimental volume is relatively small.

**Questions:**

1. The setting of the task in the article mainly starts from the high similarity continuous task, so whether the high similarity continuous learning has practical application, and whether this method can have a broad development prospect in the future technology development.
2. In the article, it is necessary to more clearly describe the task premise, between the similarity in the similarity continuous learning and the evolution in the evolutionary tasks, and then describe the clarity and rationality, why the similar continuous tasks are more consistent with the evolutionary principles.
3. In the article, I learned the role of positive learning and reverse learning in the experiment, but I want to know whether reverse learning alone is feasible, whether positive learning has an inhibitory effect on reverse learning?
4. In Figure 1 of the article, although the meaning of the expression can be roughly understood, the graph line is a little confused, especially the meaning of the black line is unclear.
5. In Figure 2 of the article, the meaning of j in the first red square and k in the second red square is not clear, and the symbols introduced in the text in Article 4.2 are inconsistent with the flow chart symbols in Figure 2, which is easy to confuse the readers' memory.


**Limitations:**

see questions

---

> ### Author Rebuttal · Authors · 2023-08-09
>
> **The assumption in the paper can better describe datasets with evolving tasks than the usual i.i.d. assumption**
> The paper's assumption describes scenarios in which the underlying distributions change as a random walk with independent increments. This type of assumption is often used to describe processes such as stock prices. The paper's assumption can better describe datasets with evolving tasks than the usual i.i.d. assumption, as described in the general response and shown in Figure 7 in Appendix G and Figure 2 in the attached pdf. We will also include other example of evolving tasks in the introduction and describe the evolving aspect of the datasets used in the experimentations, as follows.
>
> Examples of evolving tasks are the classification of portraits from different time periods [1] and of spam emails over time [41]; in these problems, the similarity between consecutive tasks (portraits and emails of consecutive time periods) is significantly higher.
>
> The proposed method is evaluated using 12 datasets composed by evolving tasks. Six datasets are formed by images with characteristics that change over time: tasks in the Yearbook dataset correspond to portraits from different years over more than a century; tasks in the ImageNet noise dataset correspond to images with increasing noise factors; tasks in the DomainNet dataset correspond to images with decreasing realism; tasks in the UTKFaces dataset correspond to face images with increasing ages; the Rotated MNIST dataset contains rotated images with increasing angles; and tasks in the CLEAR dataset correspond to images with a natural temporal evolution of visual concepts. The rest of the datasets are formed by streams of tabular data that are segmented by time: the Power Supply dataset contains power supply records over time; the Usenet datasets contain a stream of messages from different newsgroups that are sequentially presented to a user; the German dataset contains information of bank clients over time; the Spam dataset contains malicious and not malicious emails received over time; and the Covertype dataset contains cartographic information of a forest area over time.
>
> **Feasibility of backward learning**
> The proposed IMRC method allows to do backward learning alone since backward mean and MSE vectors are obtained recursively from those for the succeeding task. We obtain the backward mean and MSE vectors using forward learning recursions in retrodiction. These vectors could then be used to obtain IMRCs parameters solving the optimization problem (3). In addition, the backward ESS in Theorem 3 shows the performance improvement of the backward learning techniques. Such ESS shows that backward learning can achieve similar positive transfer as forward learning. Theorem 3 also shows that forward and backward learning results in higher ESS. Such qualitative consequences of the theoretical results are corroborated by the numerical results in Section 5 and Appendix G, for instance in Figures 4, 5, and 6. Such figures show the average classification error of single-task learning, forward learning, and forward and backward learning. Therefore, such results show that forward and backward learning improves performance in comparison with forward learning.
>
> **Additional explanations of Figure 1**
> Following the reviewer's suggestion, the updated caption of such figure will more clearly describe its contents, including the meaning of the black line. Figure~1 in the paper shows an example of evolving tasks and how the proposed techniques can adapt to evolving tasks and effectively exploit forward and backward learning. The example of evolving tasks is the classification of portraits from different decades; in this problem, the similarity between consecutive tasks is significantly higher. The black line in the figure represents the evolution of the underlying distributions that characterize different task (black dots). IMRCs minimize the worst-case error probability over uncertainty sets that can include the underlying distribution. For each task, the proposed methodology obtains a single task uncertainty set (blue hexagons) leveraging information only from the corresponding task (blue arrows), and a forward uncertainty set (green hexagons) from the forward uncertainty set for the preceding task (green arrows) and the single task uncertainty sets. Then, we obtain forward and backward uncertainty sets (red hexagons) from the forward and backward uncertainty sets for the succeeding tasks (red arrows) and the forward uncertainty sets. Such transfer of information significantly increases the ESS of each task that is represented by the size of the uncertainty set since decreasing uncertainty sets increases the ESS.
>
> **Additional explanations of symbols used in Figure 2**
> We thank the Reviewer for finding out the typo of j in the first red square and k in the second red in Figure 2. We will fix it in the new version of the paper. In addition, we will clarify the symbols used in Figure 2 and in Section 4.2. Figure 2 in the paper depicts the flow diagram for the proposed IMRC methodology for each j-th task, while the text in Section 4.2 describes the procedure for the proposed IMRC methodology at each step k. The IMRC method obtains, for each j-th task, forward mean vector $\tau_j^\rightharpoonup$ leveraging information from preceding tasks and the sample average $\tau_j$. Reciprocally, backward mean vectors $\tau\_{j+1}^{\leftharpoondown k}$ for each j-th task are obtained leveraging information from the k-th task through the sample average $\tau_k$. Then, the forward and backward mean vectors $\tau_j^{\rightleftharpoons k}$ are obtained from the forward mean vectors and the backward mean vectors. Note that the forward mean vector and sample average are obtained for each j-th task at step j. Then, at each step k, the IMRC method obtains forward mean vector for the k-th task and obtains the forward and backward mean vectors for each j-th task.

---

> > ### Comment · Reviewer_C5UC · 2023-08-18
> >
> > Thank you for the detailed feedback. I will keep my score.

---

### Official Review · Reviewer_spbD · 2023-07-06

**Soundness:** 3 good
**Presentation:** 3 good
**Contribution:** 3 good
**Rating:** 5
**Confidence:** 1

**Summary:**

IMRCs (Incremental Minimax Risk Classifiers) are a technique for incremental learning of a growing sequence of classification tasks. The key feature of IMRCs is their ability to exploit the similarity between consecutive tasks by leveraging both forward and backward learning.
Forward learning utilizes information from preceding tasks to improve the classification performance of the current task. It recursively updates the mean and mean squared error (MSE) vectors by considering the difference between the current task and the preceding task. Backward learning, on the other hand, uses information from the current task to improve the classification performance of preceding tasks.

**Strengths:**

(1) IMRCs leverage the evolving nature of tasks in a growing sequence. By utilizing both forward and backward learning, IMRCs effectively exploit the similarity between consecutive tasks.

(2) By leveraging information from preceding and succeeding tasks, IMRCs can incorporate additional knowledge and enhance classification performance.

(3) IMRCs minimize the worst-case error probabilities over uncertainty sets. This means that they provide robust classification performance, even in the face of uncertainty or variations in the underlying distributions of tasks.

**Weaknesses:**

(1) The forward and backward learning processes, along with the uncertainty set updates, can require significant computational resources and time. How about the training time compared to other methods?

(2) The selection of appropriate uncertainty sets can be challenging and may require prior knowledge or assumptions about the task distributions.

(3)  The authors assume that the distributional changes between tasks (p2-p1, p3-p2, ...) are independent and have a mean of 0. How can IMRCs be extended or modified to handle scenarios with non-zero-mean changes between consecutive tasks or explore novel tasks outside the existing sequence?

**Questions:**

See weakness part.

---

> ### Author Rebuttal · Authors · 2023-08-09
>
> **Complexity of IMRCs and running time in comparison with other methods and for different backward steps**
>
> In the final version of the paper, we will show that the running time of IMRCs is similar to other state-of-the-art techniques and increases moderately with the number of backward steps and the number of tasks, as described in the general response. Figures 1(a) and 1(d) in the attached pdf show the average running time per task for different number of backward steps and sample sizes using Yearbook and Spam datasets; and Figures 1(b), 1(c), 1(e), and 1(f) in the attached pdf show the running time for different number of backward steps, number of tasks, and sample sizes $n = 10$ and $n = 100$ using Yearbook and Spam datasets. Such figures complement those presented in Table 6 in the current Appendix G. These results show that the complexity of IMRCs increases linearly with the number of backward steps and the number of tasks in agreement with the analysis in Section 4.2.
>
> **The assumption in the paper is not stronger than the usual i.i.d. assumption**
>
> In this paper, we develop techniques designed for evolving tasks that can be mathematically modeled assuming that the changes between consecutive distributions $\mathrm{p}_{2} - \mathrm{p}_1, \mathrm{p}_3 - \mathrm{p}_2, \ldots$ are independent and zero-mean. Specifically, the assumption in the paper describes scenarios in which the underlying distributions change as a random walk with independent increments, so that it can more accurately describe evolving tasks. For instance, such type of assumption is often used to describe processes such as stock prices. The assumption in the paper differs from the usual i.i.d. assumption but is not necessarily stronger. The main difference between both assumptions is that with the random walk assumption each difference between consecutive distributions is independent of the others, while with the i.i.d. assumption each distribution $\mathrm{p}_i$ is independent of the others.
>
> **Mean of changes between consecutive distributions**
>
> IMRCs can be modified to handle scenarios with non-zero-mean changes between consecutive distributions. If $m_j$ is the mean of the change between the $j$-th and the $(j-1)$-th task, then the mean vectors evolve over time steps through the linear dynamical system
> $$\boldsymbol{\tau}_j^\infty = \boldsymbol{\tau}\_{j-1}^\infty + \boldsymbol{w}_j$$ where vectors $\boldsymbol{w}_j$ for $j \in \{2, 3, \ldots, k\}$ are independent and mean $m_j$. Then, we can obtain recursions for mean and MSE vectors after some algebra from the Kalman filter recursions and fixed-lag smoother recursions. Such recursions are given by the recursions in the paper subtracting the expectation of changes between consecutive distributions.
>
> IMRCs can be used in scenarios with an ever-increasing sequence of tasks. For each new task in the sequence, IMRCs obtain mean and MSE vectors for the new task using forward learning recursions, and then obtain mean and MSE vectors for each task in the sequence using forward and backward learning recursions.

---

> > ### Comment · Reviewer_spbD · 2023-08-18
> > **Reviewer spbD**
> >
> > Thank you for the detailed feedback. I will keep my score which leans toward acceptance.

---

### Official Review · Reviewer_V6pS · 2023-07-07

**Soundness:** 3 good
**Presentation:** 3 good
**Contribution:** 2 fair
**Rating:** 5
**Confidence:** 3

**Summary:**

This paper presents a novel method to tackle continual learning, called IMRCs which is able to exploit forward and backward learning to account for evolving tasks. The authors provide theoretical guarantees on the performance of IMRCs in terms of the tasks' expected quadratic change and the number of tasks. Authors compared the proposed method with prior work on multiple datasets, sample sizes, and number of tasks.

**Strengths:**

+  The idea seems interesting and performance guarantee in the context of continual learning is a promising research direction.
+ A thorough theoretical analysis of IMRCs is provided.

**Weaknesses:**

- My main concern about this paper is the experimental section. The diversity of the datasets and experiments with different sample sizes is a plus but at the same time it is different from the common continual learning setting in the literature and the baselines are outdated whereas the field of continual learning is rapidly growing and there has been numerous works developed between 2021 and now.

- The paper does not discuss the computational complexity of IMRCs.

**Questions:**

- Authors mention they used the original hyper parameters for the baseline. But given that some methods such as MER, GEM, EWC which were introduced on image data, cannot be taken as is and applied to tabular datasets without tuning hyper parameters. How did the authors make sure of a fair comparison? Note that results for some methods such as GEM and MER are somewhat equal to those from EWC (Table 1) but in vision applications this is not expected and EWC is always outperformed by every single memory-based method.

**Limitations:**

No.

---

> ### Author Rebuttal · Authors · 2023-08-09
>
> **Computational complexity of IMRCs**
>
> In the final version we will further discuss the computational complexity of IMRCs, the running time for different number of backward steps and number of tasks, and the running time in comparison with the state-of-the-art methods, as described in the general response.
>
> **Scope of the paper and experimental comparisons with the rapidly growing set of algorithms for continual learning**
>
> As the reviewer points out, the field of continual learning is rapidly growing. This paper aims to present methodological contributions that enable to effectively exploit forward and backward learning and account for evolving tasks. The proposed IMRCs determine classification rules minimizing worst-case error probabilities over uncertainty sets that can contain the sequence of evolving underlying distributions. Specifically, we propose forward learning techniques that recursively use the information from preceding tasks to reduce the uncertainty set of the last task, and we propose forward and backward learning techniques that use the information from the last task to further reduce the sequence of uncertainty sets obtained with forward learning. In addition, we describe the performance guarantees of the methods introduced and analytically characterize the increase in ESS provided by the presented methods in terms of the expected quadratic change between consecutive tasks and the number of tasks. The main contributions in the paper are in terms of new methodological approaches and theoretical analysis of the methods presented. The paper also includes sizeable experimental results with multiple datasets and baseline methods, but an exhaustive experimental comparison that include the most recent techniques is beyond the scope of the paper.
>
> The existing techniques for concept drift adaptation are designed for evolving tasks but only aim to learn the last task in the sequence, while the existing techniques for continual learning aim to learn the whole sequence of tasks but are designed for situations where tasks’ similarities do not depend on the time steps when tasks are observed. The methodology proposed fills this gap in the existing literature and present techniques that simultaneously learn a sequence of tasks and account for evolving tasks.
>
> **Hyper-parameters for the baselines and comparison methods**
>
> As described above, the main goal of the paper is to present methodological contributions that enable to learn a sequence of evolving tasks; rather than improving performance in comparison with the most recent state-of-the-art methods in each specific dataset. While hyper-parameter fine-tuning is a crucial step in achieving the best possible results, it often involves a significant amount of computational resources and extensive experimentation. Given the focus on methodological contribution, we demonstrate the effectiveness of the techniques introduced by using default hyper-parameters for all datasets, methods, and sample sizes.
>
> EWC is outperformed by GEM and MER in most of vision datasets such as Yearbook, I. noise, DomainNet, R. MNIST, and CLEAR datasets. For the tabular datasets, results for GEM and MER are somewhat equal to those from EWC. This fact can be due to the strong similarity between consecutive tasks in later datasets that have been often used as benchmarks for concept drift adaptation. Note that GEM, MER, and EWC are designed for situations where tasks’ similarities do not depend on the time steps when tasks are observed so that their performance is expected to be similarly deficient in situations where the tasks' similarities between consecutive tasks are markedly higher.

---

> > ### Comment · Reviewer_V6pS · 2023-08-17
> >
> > I would like to thank the authors for their thorough rebuttal. I appreciate the effort taken to provide clarity on the computational complexity aspect. However, I still have reservations regarding the experimental section. While I concur with the authors' assessment of the paper's contributions, I believe that the experimental design should align more closely with these contributions. Specifically, I find it important to address the rationale behind EWC, a regularization technique with inherent limitations, performing at a comparable level to memory-based methods. The latter possess greater flexibility in retaining past knowledge due to their capacity to store samples. The authors' justification, centered around the "strong similarity between consecutive tasks in later datasets," suggests that forgetting might not be a significant challenge in the selected datasets. Consequently, even a less potent method like EWC could exhibit satisfactory results. Thus, I question the reliability of such datasets as suitable benchmarks. Furthermore, I respectfully disagree with the statement that "GEM, MER, and EWC are designed for situations where tasks’ similarities do not depend on the time steps" being a fair reason for their comparable performance. These methods have undergone testing on continual learning datasets with temporal disparities in the past, and the outcome trends have persisted.
> >
> > In conclusion, after thoroughly reviewing both the critiques and responses, I am inclined to revise my initial assessment. I acknowledge the strengths highlighted by myself and other reviewers. As a result, I am willing to adjust my score accordingly and raise it from 3 to 5.

---

> > > ### Author Response · Authors · 2023-08-18
> > >
> > > We would like to thank the reviewer for his/her careful reading of the responses in the rebuttal and the appreciation of the paper's main contributions. In addition, we would like to thank the reviewer for raising the score from 3 to 5.
> > > In the final version of the paper, we will clarify that the main aim of the paper is to present methodological contributions as well as the theoretical properties of the methods proposed.

---

### Official Review · Reviewer_3qiG · 2023-07-26

**Soundness:** 4 excellent
**Presentation:** 3 good
**Contribution:** 3 good
**Rating:** 7
**Confidence:** 4

**Summary:**

The papers studies a specific case of continual learning over a sequence of tasks. Authors extend the typical i.i.d. assumption of task meta-distribution to the case where only the differences between subsequent task distributions are assumed to be independent and zero-mean. Under this assumption, they devise two versions of a minimax risk classifier and prove performance guarantees that improve upon single-task bounds by relying on effective sample size instead of just the number of samples in a task. The effective sample size is no smaller than the single task size and can be larger if the quadratic change between consecutive tasks behaves in a favorable way, e.g. decreasing over time. The authors also present an experimental study of the proposed algorithms and show that it either improves the baseline performance or achieves similar one across 12 datasets.

**Strengths:**

### Originality
This is a novel approach that relies on independent differences between subsequent task distributions rather than on the typical i.i.d. assumption.

### Quality
* Paper studies the presented algorithm theoretically and experimentally.
* Illustrative theoretical guarantees that are always no worse than a single task with a clear influence of the task distribution properties in the form of effective sample size.
* Experiments uses 12 different datasets and cover 7 baselines.
* Authors provide insights into impact of number of tasks and number of samples on the performance of different variants of the algorithm.

### Clarity
The paper is well written and easy to follow in general.

### Significance
There are numerous ways of how i.i.d. assumption can be relaxed and paper studies one of them. The significance will rely on how close the new assumption is to practical scenarios. From experimental setup it seems to be a rather specific case that relies too much on exact task ordering, thus I would expect the paper to have average impact from theoretical point of view, but quite limited impact in practice.

**Weaknesses:**

The paper doesn’t have major weaknesses. There are things that can be improved in the presentation and discussing limitations, which I cover in questions section.

**Questions:**

The following things would help with the presentation:
* The notations used are overloaded (e.g. by using double superscripts)  and sometimes are hard to follow. The paper would benefit from a small table with all notations in one place.
* Experiments present the result for samples size of 10 without any additional comments on exact value of 10. Appendix shows that the general conclusion is valid irrespective of the sample size, so it would be good to highlight that.
* The main assumption on the task distribution needs to be made more precise and explicit in the theorems (introduce assumptions separately and then refer to it in theorems). A discussion on how realistic it is and providing more than one example would go a long way.
* Theoretical analysis needs a discussion on different regimes where the ESS improvements would be realised and what they mean for the task distributions process. Authors do have a short discussion of that after Theorem 4, but it lacks the connection between different regimes of nd^2 and process properties. For example, $nd^2 \leq \frac{1}{j^2}$ translates into a rather fast convergence requirement of the task distribution process.
* It would be great to discuss what happens to the algorithm when the main assumption breaks. Does it go back to single task performance? How would one figure this out in practice?

**Limitations:**

The limitation of the approach are not adequantly discussed. I’ve made a few suggestion in the questions section on how to address that.

---

> ### Author Rebuttal · Authors · 2023-08-09
>
> **Table with main notations used in the paper**
>
> In the final version of the paper we will include Table 2 in the attached pdf that shows the main notations used. Specifically, such table shows the notation used for the mean vector, the confidence vector, the MSE vector, the ESS, the uncertainty set, and the minimax risk. In addition, symbols without superscript represent variables obtained using single task learning; symbols with right harpoon superscript represent variables obtained using forward learning; symbols with left harpoon superscript represent variables obtained using backward learning; and symbols with right left harpoon superscript represent variables obtained using forward and backward learning. Due to space constraints, we will include the table in the appendices and will reference it in the Notation paragraph.
>
> **IMRCs are successful irrespective of the sample size**
>
> We will follow the reviewer's suggestion and highlight that the improved performance of IMRCs compared to the state-of-the-art techniques is similar irrespective of the sample size. We will clarify that the results for $n = 10$ samples per task in Table 1 in the main paper are complemented with results for $n = 50, 100$, and $150$ samples per task in Tables 4, 5, and 6 in Appendix G. These results show that the improved performance of IMRCs compared to the state-of-the-art techniques is similar for different sample sizes.
>
> **More formally describe the paper's assumption and reference to it in theorems**
>
> We will follow the reviewer's suggestion and more formally describe the assumption in Section 2, as follows. Most existing continual learning techniques are designed for tasks characterized by distributions $\mathrm{p}_1, \mathrm{p}_2, ...$ such that the tasks’ distributions $\mathrm{p}_i$ are independent and identically distributed (i.i.d.) for $i = 1, 2, ...$. In the following, we propose techniques designed for evolving tasks that are characterized by distributions $\mathrm{p}_1, \mathrm{p}_2, ...$ such that the changes between consecutive distributions $\mathrm{p}\_{i+1} - \mathrm{p}_i$ are independent and zero-mean for $i = 1, 2, ..$. In addition, we will refer to such assumption as the ''evolving tasks assumption'' in Section 2 and in theorems.
>
> **Suitability of the assumption in the paper for evolving tasks and additional examples**
>
> The assumption in the paper describes scenarios in which the underlying distributions change as a random walk with independent increments. This type of assumption is often used to describe processes such as stock prices. The assumption in the paper can better describe real-world datasets with evolving tasks than the usual i.i.d. assumption, as described in the general response and shown in Figure 7 in the current Appendix G and Figure 2 in the attached pdf. We will also include other example of evolving tasks in the introduction and describe the evolving aspect of the datasets used in the experimentations, as follows.
>
> Examples of evolving tasks are the classification of portraits from different time periods [1] and the classification of spam emails over time [41]; in these problems, the similarity between consecutive tasks (portraits of consecutive time periods and emails from consecutive years) is significantly higher.
>
> The proposed method is evaluated using 12 datasets composed by evolving tasks  [9, 1, 34–41]. Six datasets are formed by images with characteristics/quality/realism that change over time: tasks in the Yearbook dataset correspond to portraits from different years over more than a century; tasks in the ImageNet noise dataset correspond to images with increasing noise factors; tasks in the DomainNet dataset correspond to images with decreasing realism; tasks in the UTKFaces dataset correspond to face images with increasing ages; the Rotated MNIST dataset contains rotated images with increasing angles; and tasks in the CLEAR dataset correspond to images with a natural temporal evolution of visual concepts. The rest of the datasets are formed by streams of tabular data that are segmented by time: the Power Supply dataset contains power supply records over time; the Usenet datasets contain a stream of messages from different newsgroups that are sequentially presented to a user; the German dataset contains information of bank clients over time; the Spam dataset contains malicious and not malicious emails received over time; and the Covertype dataset contains cartographic information of a forest area over time.
>
> **Discussion on regimes of the ESS improvements**
>
> In the updated version of the paper we will further discuss the regimes of the ESS improvements shown in Theorem 4 and Figure 3, as follows. The increase in ESS can be classified into three regimes depending on the relationship between the task index $j$, the sample size $n$, and the expected quadratic change $d^2$. The ESS becomes proportional to the total number of tasks $k$ if $nd^2$ is rather small (very small sample sizes or very slow changes in the distribution); the ESS quickly increases when $nd^2$ becomes small (reduced sample sizes and moderate changes in the distribution); and the ESS is only marginaly larger than the sample size for sizeble values of $nd^2$ (large samples sizes or drastic changes in the distribution).
>
> **IMRCs' behavior when the similarity between consecutive tasks is small**
>
> IMRCs are designed for situations where consecutive tasks are significantly more similar and the expected quadratic change ($d^2$) between consecutive tasks is small. If, instead, this expected quadratic change is large we would have that the forward and forward and backward mean vectors in equations (6) and (10), respectively, become the mean vector of single task learning, as hinted by the reviewer. Such cases can be detected in practice because the estimate for $d^2$ in equation (7) would become large. Then, in these cases, the performance of IMRCs becomes the performance of single task learning.

---

> > ### Comment · Reviewer_3qiG · 2023-08-16
> >
> > Thank you for the detailed rebuttal. You have addressed most of my questions/concerns expect for the last one: "It would be great to discuss what happens to the algorithm when the main assumption breaks. Does it go back to single task performance? How would one figure this out in practice?"
> >
> > I probably didn't express myself well enough. What I meant is adding a discussion on limitations of your main assumption. When will your approach stop working? Main assumption "breaks" when consecutive distributions stop being independent and/or zero-mean. You have clarified how to adapt the approach for the non-zero mean in one of other rebuttals, but what would happen when the distributions are dependent? Obviously your theoretical bounds won't hold, but would approach reduce to learning each task independently? Paper would definitely benfit from this information.
> >
> > Overall, after reading the rebuttal and other reviews, my recommendation remains the same.

---

> > > ### Author Response · Authors · 2023-08-17
> > >
> > > We would like to thank the reviewer for his/her careful reading of the responses in the rebuttal and the positive feedback provided that will help us describe the methods' reliance on the assumptions.
> > >
> > > The proposed method should improve performance as long as consecutive tasks have a higher similarity (evolving), even if their differences are dependent. IMRCs obtain mean vectors using those for the consecutive tasks and the expected quadratic change between consecutive tasks $\boldsymbol{d}_j^2$. If $\boldsymbol{d}_j^2$ is large, we would have that the mean vectors become the mean vectors of single task learning; while if $\boldsymbol{d}_j^2$ is small, we would have that the mean vectors become the mean vectors of standard supervised classification. The main limitation of the methods proposed is that IMRCs cannot exploit strong similarities among non-consecutive tasks. The methods presented are designed for situations in which tasks are evolving in the sense that consecutive tasks often have a higher similarity.

---

### Official Review · Reviewer_qXgD · 2023-07-30

**Soundness:** 4 excellent
**Presentation:** 3 good
**Contribution:** 4 excellent
**Rating:** 6
**Confidence:** 3

**Summary:**

Though class-incremental learning has been studied the most as a default setting for continual learning, task-incremental study could be crucial for other areas. This paper focuses particularly on the case where tasks being introduced over time are interrelated under specific conditions. Under the supervised learning framework, the task distributions $p_t$ (of instances and labels at any time t) are independent and identically distributed, which would be a strong and less-realistic condition. In contrast, this paper covers the case where the difference of two consecutive task distributions is independent, making the difference between two distant tasks have increasing variance over time. Then the authors propose Incremental Risk Minimization Classifier (IMRC) method that effectively leverage forward and backward learning algorithms, improving the performance against existing methods on various datasets, sample sizes, and the number of tasks.

**Strengths:**

(1)	The paper formally defines a non-trivial task-incremental learning setting whose assumption is more realistic than the earlier approach.

(2)	The paper analytically shows performance guarantees and effective sample sizes as well as forward and backward learning algorithm.

(3)	The paper addresses empirical improvement on multiple datasets against existing state-of-the-art algorithms.


**Weaknesses:**

(1)	Current draft is rarely self-contained. Readers must read the appendix frequently. Minimally, having explanations about methods and datasets in Table 1 would significantly increase the readability.

(2)	Run-time trade-off of considering the backward (possibly per different sample sizes) are missing.

(3)	Characteristics of each experimental dataset: how much they agree and disagree the independence assumption across consecutive task distributions are missing.



**Questions:**

(1)	Providing comprehensive details about different tasks and their distinctive characteristics helps readers better understanding the contributions of the paper beyond the theoretical excitement. Formalizing the time-evolving task with an independence assumption on consecutive task difference sounds exceptional. It will be great to know how much this assumption is agreed and violated on individual tasks on experiments.

(2)	In addition to (1), accessing real examples or qualitative explanations about hard cases (that were not successful in the previous approaches, but exceptionally successful with the current IMRC) will be useful. For example, it seems IMRC is distinguished notably from other algorithms on P. Supply, Usenet1, and Spam datasets. Understanding the rationale will be insightful.

(3)	Most importantly, run-time trade-off for doing backward process must be measured for making the provable guarantees practically effective. As the trade-off would change for different number of tasks and sample sizes, practitioners require better guidance.


**Limitations:**

No specific limitations are described or probed.

---

> ### Author Rebuttal · Authors · 2023-08-09
>
> **Explanations about methods and datasets**
>
> In the final version of the paper, we will further describe the methods and datasets used in Table 1 to make the main paper more self-contained as suggested by the reviewer. Specifically, we will include the following comments on the datasets and methods.
>
> The proposed method is evaluated using 12 public datasets composed by evolving tasks  [9, 1, 34–41]. Six datasets are formed by images with characteristics/quality/realism that change over time: tasks in the Yearbook dataset correspond to portraits from different years over more than a century; tasks in the ImageNet noise dataset correspond to images with increasing noise factors; tasks in the DomainNet dataset correspond to images with decreasing realism; tasks in the UTKFaces dataset correspond to face images with increasing ages; the Rotated MNIST dataset contains rotated images with increasing angles; and tasks in the CLEAR dataset correspond to images with a natural temporal evolution of visual concepts. The rest of the datasets are formed by streams of tabular data that are segmented by time: the Power Supply dataset contains power supply records over time; the Usenet datasets contain a stream of messages from different newsgroups that are sequentially presented to a user; the German dataset contains information of bank clients over time; the Spam dataset contains malicious and not malicious emails received over time; and the Covertype dataset contains cartographic information of a forest area over time.
>
> In Section 5, we compare the results of IMRC methods with 7 state-of-the-art-techniques [4, 16, 3, 5, 8, 9, 15]: GEM method [4] is a technique developed for continual learning based on experience replay and learns each new task using a stochastic gradient descent with inequality constraints given by the losses of preceding tasks; MER method [16] is a technique developed for continual learning based on experience replay and learns each new task using sample sets that include stored samples from preceding tasks; EWC method [5] is a technique developed for continual learning based on regularization and learns each new task using regularization parameters based on the relevance for preceding tasks; ELLA method [3] is a technique developed for continual learning based on dynamic architectures and learns each new task transferring knowledge from a shared basis of task models; Condor method [8] is a technique developed for concept drift adaptation based on weight factors and adapts to evolving tasks by weighting the models in an ensemble at each time step; AUE method [15] is a technique developed for concept drift adaptation based on weight factors and adapts to evolving tasks by incrementally updating all classifiers in an ensemble and weighting them with non-linear error functions; and DriftSurf method [9] is a technique developed for concept drift adaptation based on sliding windows and adapts to evolving tasks by restarting the model when a change in the distribution is detected.
>
> **The assumption in the paper more accurate describes evolving tasks than the usual i.i.d. assumption**
>
> We will show that the assumption in the paper can better describe datasets with evolving tasks than the usual i.i.d. assumption as described in the general response. Specifically, Figure 2 in the attached pdf shows the averaged partial autocorrelation of the mean vectors components +/- their standard deviations for different lags using Power Supply, Covertype, and R. MNIST datasets. Such results complement those using Yearbook and UTKFaces datasets presented in Figure 7 in the current Appendix G. Figure 7 in Appendix G and Figure 2 in the attached pdf clearly show that the paper's assumption better describes evolving tasks than the usual i.i.d. assumption. Note that in the i.i.d. case the partial autocorrelation at lag 1 would be 0, while such figures show a partial autocorrelation clearly non-zero at lag 1.
>
> **IMRCs are especially successful in Power Supply, Usenet1, and Spam datasets**
>
> As the reviewer points out, IMRCs significantly improve performance in comparison with other algorithms on Power Supply, Usenet1, and Spam datasets. Such datasets are used as benchmarks in concept drift adaptation [9, 39, 41]  and have a markedly strong similarity between consecutive tasks. As described in the first response, the Power Supply dataset contains power supply records over time; the Usenet1 dataset contains a stream of messages from different news-groups that are sequentially presented to a user; and the Spam dataset contains malicious and not malicious emails received over time. Existing techniques designed for continual learning do not account for evolving tasks, and existing techniques designed for concept drift adaptation do not exploit backward learning. IMRCs are successful in the datasets cited by the reviewer because the methods proposed account for evolving tasks and effectively exploit forward and backward learning.
>
> **Running times for different backward steps, number of tasks, and sample sizes**
>
> The running time of IMRCs increases moderately with the number of backward steps and the number of tasks as described in the general response. Figures 1(a) and 1(d) in the attached pdf show the average running time per task for different numbers of backward steps and sample sizes using Yearbook and Spam datasets; and Figures 1(b), 1(c), 1(e), and 1(f) in the attached pdf show the running time for different numbers of backward steps, number of tasks, and sample sizes $n = 10$ and $n = 100$ using Yearbook and Spam datasets. Such results complement those presented in Table 6 in the current Appendix G. These results and the analysis in Section 4.2 show that the complexity of IMRCs increases linearly with the number of backward steps and the number of tasks.

---

> > ### Comment · Reviewer_qXgD · 2023-08-18
> > **Thanks for your rebuttal.**
> >
> > Thanks the authors for their feedback on my comments and suggestions. I will keep my score based on their clarifiations.

---

### Author Rebuttal · Authors · 2023-08-09

In the responses below we are confident to have addressed all the comments and questions made by the reviewers. Please let us know if you have any additional inquiry so that we can completely clarify any aspect in the paper during the rebuttal period.

We plan to use the extra page allowed to clarify the questions raised by the reviewers as described in the responses below. In this general response, we clarify the two main questions raised by the reviewers regarding the running time of the methods proposed and the assumption for evolving tasks.

**Running times in comparison with other methods and for different backward steps**

Table 6 in Appendix G and Table 1 in the attached pdf show that the running time of the IMRC method is similar to other state-of-the-art methods, and Table 6 in Appendix G and Figure 1 in the attached pdf show that the running time of IMRCs increases moderately with the number of backward steps. Specifically, Table 1 in the attached pdf shows the average running time per task of IMRCs in comparison with the state-of-the-art methods and Figure 1 in the attached pdf shows that the complexity increases linearly with the number of backward steps and the number of tasks. Such results complement those presented in Table 6 in the current Appendix G and agree with the theoretical analysis in Section 4.2. This analysis shows that, for $k$ steps, IMRCs have computational complexity $\mathcal{O}((b+1)Kmk)$ and memory complexity $\mathcal{O}((b+k)m)$ where $K$ is the number of iterations used for the convex optimization problem~(3), $m$ is the length of the feature vector, and $b$ is the number of backward steps.

**The assumption in the paper more accurate describes evolving tasks than the usual i.i.d. assumption**

Figure 7 in Appendix G and Figure 2 in the attached pdf show that the assumption in the paper can better describe datasets with evolving tasks than the usual i.i.d. assumption. We evaluate the paper's assumption of change between consecutive tasks being independent and zero-mean by assessing the partial autocorrelation of mean vectors. Partial autocorrelations are the usual tool to assess if a process is a random walk (see Section 4 in Cowpertwait and Andrew V. Metcalfe (2009)). In particular, the partial autocorrelation at any lag would be zero if tasks are i.i.d.; while the partial autocorrelation at lag 1 is larger than zero if tasks satisfy the assumption of Section 2. Figure 7 in the current Appendix G and Figure 2 in the attached pdf show the averaged partial autocorrelation of the mean vectors components +/- their standard deviations for different lags. Such figures show a partial autocorrelation clearly non-zero at lag 1 that reflects a dependence between consecutive mean vectors in accordance with the assumption of Section~2.

Paul SP Cowpertwait and Andrew V. Metcalfe. Introductory time series with R. Springer Science \& Business Media, 2009.

---

### Decision · Program_Chairs · 2023-09-21

**Decision:**

Accept (poster)

**Comment:**

All reviewers found the work interesting and relevant. After the author’s response, the reviewers were mostly satisfied and did not see the need for further clarifications. Ultimately, all reviewers recommended acceptance.